# Exploring the role of different data types and timescales for the quality of marine biogeochemical model calibration

Iris Kriest[1], Julia Getzlaff[1], Angela Landolfi[2], Volkmar Sauerland[1], Markus Schartau[1], and Andreas Oschlies[1]

[1]GEOMAR Helmholtz-Zentrum für Ozeanforschung Kiel, Düsternbrooker Weg 20, D-24105 Kiel, Germany
[2]ISMAR-CNR, via Fosso del cavaliere 100, 00133 Rome, Italy

**Correspondence:** Iris Kriest (ikriest@geomar.de)

**Abstract.**

Global biogeochemical ocean models help to investigate the present and potential future state of the ocean, its productivity and cascading effects on higher trophic levels such as fish. They are often subjectively tuned against data sets of inorganic tracers and surface chlorophyll and only very rarely against organic components such as particulate organic carbon or zooplankton.
The resulting uncertainty in biogeochemical model parameters (and parameterisations) associated with these components can explain some of the large spread of global model solutions with regard to the cycling of organic matter and its impacts on biogeochemical tracer distributions, such as oxygen minimum zones (OMZs). A second source of uncertainty arises from differences in the model spin-up length, as, so far, there seems to be no agreement on the required simulation time that should elapse before a global model is assessed against observations.

We investigated these two sources of uncertainty by optimising a global biogeochemical ocean model against the root-mean-squared error (RMSE) of six different combinations of data sets and different spin-up times. Besides nutrients and oxygen, the observational data sets also included phyto- and zooplankton, as well as dissolved and particulate organic phosphorus (DOP and POP, respectively). We further analysed the optimised model performance with regard to global biogeochemical fluxes, oxygen inventory and OMZ volume.

Following the optimisation procedure we evaluated the RMSE for all tracers located in the upper 100 m (except for POP, for which we considered the entire vertical domain), regardless of their consideration during optimisation. For the different optimal model solutions we find a narrow range of the RMSE, between 14% of the average RMSE after 10 years and 24% after 3000 years of simulation. Global biogeochemical fluxes, global oxygen bias and OMZ volume showed a much stronger divergence among the models and over time than RMSE, indicating that even models that are similar with regard to local surface tracer concentrations can perform very differently when assessed against the global diagnostics for oxygen. Considering organic tracers in the optimisation had a strong impact on the particle flux exponent ("Martin $b$") and may reduce much of the uncertainty in this parameter and the resulting deep particle flux. Independent of the optimisation setup, the OMZ volume showed a particularly sensitive response with strong trends over time even after 3000 years of simulation time (despite the constant physical forcing), a high sensitivity to simulation time, as well as the highest sensitivity to model parameters arising
from the tuning strategy setup (variation of almost 80% of the ensemble mean).

In conclusion, calibration against observations of organic tracers can help to improve global biogeochemical models even after short spin-up times; here, especially observations of deep particle flux could provide a powerful constraint. However, a large uncertainty remains with regard to global OMZ volume and its evolution over time, which can show a very dynamic behaviour during the model spin-up, that renders temporal extrapolation to a final 'equilibrium' state difficult, if not impossible. Given that the real ocean shows variations on many timescales, the assumption of observations representing a steady-state ocean may require some reconsideration.

## 1 Introduction

Global biogeochemical ocean models, especially when combined with data assimilation techniques, serve as useful tools to generate spatially and temporally consistent global fields of dissolved and particulate ocean tracers, such as nutrients, oxygen, or organic constituents, from sparse observations. More important, when embedded in Earth system models, they can be used to investigate the present state of the ocean and its biogeochemistry, including its productivity (e.g. Kwiatkowski et al., 2017), up to higher trophic levels (e.g. Chust et al., 2014; Stock et al., 2014) and fish (Galbraith et al., 2017; Mullon et al., 2017; Stock et al., 2017) as well as its sensitivity to a changing climate. Besides direct effects of productivity on fish production, complex feedback processes, in particular when leading to ocean deoxygenation and expanding oxygen minimum zones (OMZs), are likely to impact on stock and recruitment of fish. For example, the amount of organic matter produced at the surface, exported to and recycled within the mesopelagic zone (about 200-1000 m) can have significant effects on OMZs. Complex interactions between different biogeochemical components can lead to large uncertainties with regard to the location and extent of OMZs simulated by global models (Cabre et al., 2015; Kriest and Oschlies, 2015), which may hamper our ability to reproduce and project the habitat of commercially relevant, oxygen-sensitive fish species (Stramma et al., 2012).

Relevance and prospects of marine biogeochemical model applications seem obvious, but challenges remain in finding unambiguous solutions of the global distribution and flux of mass, and in quantifying uncertainties of respective model estimates (Schartau et al., 2017). The choice of values assigned to the biogeochemical model parameters is known to have considerable effects on model performance (Kriest et al., 2010, 2012). Albeit undisputed, problems of parameter identification remain underrated, and sensitivity analyses of the entire parameter space are the exception rather than the rule, even at local or regional scales (Arhonditsis and Brett, 2004; Leles et al., 2016). Likewise, a comprehensive analysis of model performance with regard to all simulated state variables is not always carried out. This bears the risk that during calibration one simulated tracer is improved at the cost of an unconstrained one (as, for example, shown in Kriest, 2017); a problem known as "calibration bias" (Arhonditsis and Brett, 2004). For example, almost two decades ago, far less than half of the studies reviewed by Arhonditsis and Brett (2004) reported performance statistics for all simulated state variables. If narrowed down to global biogeochemical model applications, validation becomes more difficult, due to the sparsity and type of data available at a global scale: while the global coverage of concentration measurements of nutrients or oxygen is relatively good, observations of plankton and organic matter are less abundant. While most of the models applied in CMIP5 and CMIP6 have been evaluated with regard to dissolved

inorganic tracers (e.g., Seferian et al., 2020), analysis of model skill especially with regard to higher trophic levels is carried out less often (but see, for example Petrik et al., 2022, who examined CMIP6 models with regard to mesozooplankton).

During model calibration, the calibration bias can have significant effects in optimal parameter estimates, and may affect model performance with regard to simulated biogeochemical fluxes such as primary and secondary production. For instance, amongst different model configurations that yield equally good fits to global nutrients and oxygen concentrations (with differences between root mean square errors $\Delta RMSE \leq 6\%$), Kriest et al. (2017) revealed significant differences in primary production (17%) and grazing (84%). Also, different tuning approaches might explain some of the spread in global primary
production found in global model inter-comparisons (Bopp et al., 2013; Kwiatkowski et al., 2014). When primary (and sometimes secondary or export) production simulated by biogeochemical (BGC) models is used to estimate present and future stocks and production of higher trophic levels (HTL) or fish (Galbraith et al., 2017; Mullon et al., 2017; Stock et al., 2017), it is likely that BGC model uncertainties will propagate into HTL estimates.

Another important aspect of model uncertainty relates to the models' spin-up time. Global models are started from some
observed or assumed distributions of simulated tracers. However, their inherent assumptions, as expressed through the model components, equations, constants, forcing and boundary conditions, may diverge from the "real world". When the model is simulated forward in time, the inherent assumptions will translate into a specific simulated tracer distribution, and will likely diverge from the observations (unless we have a perfect model). The discrepancy between model and observations will change over simulation (spin-up) time, until finally, and under climatological, seasonally varying forcing, the model reaches
an equilibrium or "steady-state". This equilibrium is characterised by a steadily repeating annual cycle, in which the tracer concentrations at all locations change from year to year only by a negligible amount. In equilibrium the model output then reflects only the assumptions that went into the model, and is usually independent of the initial tracer distribution and, in case of tracer exchange through open boundaries, the initial tracer inventory.

Because of the slow ocean overturning circulation, it requires millennia of numerical integration to reach an equilibrated
biogeochemical state on a global scale (Wunsch and Heimbach, 2008). The length of spin-up time depends on the upper boundary condition (e.g., Wunsch and Heimbach, 2008; Primeau and Deleersnijder, 2009; Siberlin and Wunsch, 2011), the lower and lateral boundary condition (e.g., Roth et al., 2014) and the tracer considered; for example, tracers with distant sources and sinks (such as nitrogen) may require longer time scales to reach equilibrium than tracers that exchange quickly with the atmospheric boundary (such as oxygen; Kriest and Oschlies, 2015). However, given the high computational demand
of global models, we find a wide variety of model spin-up times, from decades (e.g. Dietze and Loeptien, 2013; Kwiatkowski et al., 2014; Henson et al., 2015; Le Quere et al., 2016) up to centuries and even millennia (e.g. 200-12.000 years; Seferian et al., 2016); sometimes model parameters are adjusted during spin-up (Lindsay et al., 2014). Spin-up times of most recent versions of CMIP6 ocean models span a wide range, from 150 years up to 12.000 years, an even wider range than those applied earlier during CMIP5 (500 to 11.900 years Seferian et al., 2020). Given the wide range of model spin-up times, we may find
some unwanted impact on model ranking in inter-comparison studies (Seferian et al., 2016). While a short-term simulation and calibration effort may provide valuable insight to plankton dynamics and may help constraining predominant seasonal variations at the sea surface (Doney et al., 2009; Le Quere et al., 2016), inferences with respect to long-term, large-scale

changes in the deep ocean remain unwarranted: For example, a model setup, that yields good model performance after a few decades or centuries of simulation can give rise to unfavourable model results with regard to large scale tracer distributions and
inventories on millennial timescales (Seferian et al., 2016; Kriest and Oschlies, 2015).

We here investigate driving factors that contribute to potential biases in parameter tuning and analyse their effect on global model performance with regard to several combinations of metrics, parameters to be calibrated, and simulation time scales. Initially, we optimised four parameters that determine the turnover of organic matter in the euphotic zone of a global bio-geochemical ocean model and two parameters related to oxygen consumption and particle flux (with two different ranges of
potential parameter values). The choice of the six parameters was partly motivated by the results from earlier optimisations (Kriest et al., 2017, 2020); we also aimed at every parameter affecting directly at least one of the biogeochemical state variables simulated by the model. This initial optimisation procedure is constrained by surface observations of all of our model's biogeochemical tracers, namely phosphate, nitrate, oxygen, and dissolved organic matter (DOM), as well as observational counterparts for particulate organic components (phytoplankton, zooplankton and detritus). To account for the uncertainty of
the particle flux parameter $b$, which can have a large influence on large-scale nutrient distribution (e.g., Kwon and Primeau, 2006; Kriest et al., 2012), this initial optimisation setup against the full data set was applied with two different potential ranges of $b$.

By including all simulated tracers in the misfit function in the two initial optimisations we aim to avoid the calibration bias mentioned above. In particular, we obviate any tendency of the optimisation procedure to reduce misfits in inorganic tracer
concentrations to the disadvantage of (otherwise unconstrained) organic components, such as DOM or plankton biomass. In three further experiments we then successively omitted observations of organic constituents and reduced the numbers of parameters to be optimised, thereby investigating the impact of different data types on optimisation performance.

Because especially changes in plankton parameters affect model performance mainly at the surface, which adjusts on timescales of years (see also Le Quere et al., 2016), these five optimisations were carried out after a spin-up of only 10
years. A final optimisation then applies a spin-up of 3000 years. To account for the unresolved effects on deep tracer concentrations over long time scales, we also investigate how the optimal model solutions of the short-term optimisations perform globally after 3000 years, when the effects of model parameters are propagated to the deep ocean via the large-scale ocean overturning circulation. This examines whether the model solutions that perform "best" at the ocean surface on decadal time scales are also appropriate in the context of longer time and larger spatial scales. In our analysis of optimised model result we
not only evaluate the tracer residuals, but also examine the effect of calibration on global biogeochemical tracer fluxes, oxygen inventory and OMZ volume, as potentially important interfaces to higher trophic levels.

## 2 Model structure and optimisations

### 2.1 Ocean biogeochemical model

All model simulations and optimisations apply the Transport Matrix Method (TMM; Khatiwala, 2007, 2018), which represents
the joint effects of advection and mixing in the form of monthly mean transport matrices (TMs) derived from a circulation field

of the Estimating the Circulation and Climate of the Ocean (ECCO) project. ECCO provides circulation fields that yield a best fit to hydrographic and remote sensing observations over the 10-year period 1992 through 2001 with a horizontal resolution of $1° \times 1°$ and 23 levels in the vertical (Stammer et al., 2004). Monthly mean wind speed, temperature and salinity of the same model are used to compute air-sea gas exchange of oxygen and temperature-dependent growth of phytoplankton and cyanobacteria.

The biogeochemical model describes the cycling of phosphorus, nitrogen and oxygen in a stoichiometrically consistent manner (MOPS -"Model of Oceanic Pelagic Stoichiometry"; Kriest and Oschlies, 2015). The model contains seven components, of which five are calculated in phosphorus units, namely phytoplankton, zooplankton, detritus, dissolved organic phosphorus and phosphate. Additionally, nitrate and oxygen are simulated, with biogeochemical interactions among the different elements coupled via fixed stoichiometric ratios. In contrast to the fixed phosphorus inventory, total nitrogen can be altered in response to variations in denitrification and nitrogen fixation. Likewise, the oxygen inventory may change according to variations in model parameters, in combination with air-sea gas exchange and circulation. Details of the model can be found in Kriest and Oschlies (2015), and the initial calibration of the model is optimisation "ECCO*" described in Kriest et al. (2020), which serves as starting point for the optimisations presented here.

## 2.2 Data sets for optimisation

To assess model skill for all seven simulated tracers we have compiled a data set of corresponding observations as detailed in Appendix A1. Similar to the optimisations by Kriest et al. (2017) and Kriest et al. (2020), simulated nutrients and oxygen are assessed against the objectively analysed data of Garcia et al. (2006a) and Garcia et al. (2006b). For phytoplankton we use surface chlorophyll data derived from remote sensing (Malin, 2013), converted to phytoplankton phosphorus by applying the algorithm by Sathyendranath et al. (2009) while assuming a fixed molar carbon-to-phosphorus ratio of 122 mol C:mol P. We note that this data set provides a quasi-synoptic global data coverage, but only for the uppermost layer of the ocean; furthermore, it includes the above-mentioned assumptions about the Chl:C ratio of chlorophyll. The simulated zooplankton biomass is evaluated against annual averages, derived from monthly mesozooplankton biomass data provided by Moriarty and O'Brien (2013). According to a preceding analysis at the few locations where both micro- and mesozooplankton data are available, annual averages of mesozooplankton biomass are similar to those of microzooplankton biomass (see Appendix A1). Therefore we assume the simulated bulk zooplankton biomass can be compared against mesozooplankton biomass observations that are multiplied by a factor of two. Because there is no direct observational equivalent to simulated detritus, we assume that simulated phytoplankton, half of the simulated zooplankton biomass (considered as the less motile microzooplankton component) and detritus contribute to particulate organic matter (POM), and compare this quantity to the data set by Martiny et al. (2014). Because this biogeochemical component can sink quite rapidly into the deep ocean, in contrast to the other data types we here consider the entire vertical domain. Finally, observations of DOP were compiled from various sources, covering locations in the Atlantic, Pacific and Indian Ocean (see Appendix A1).

We note that in contrast to the inorganic tracers and phytoplankton the data sets for organic components (in particular: zooplankton, POP and DOP) are much more sparse in space and time (see also Appendix A1 and Table A1). They are typically

sampled and measured during ship cruises, and provide rather a local snapshot of the biogeochemical system, that is likely affected by the local physical conditions. The sparseness and episodic nature of the observations, in conjunction with the coarse and climatological global model circulation can pose some limitations for the assessment of model skill against these tracers, which are discussed further below. In this first attempt to calibrate a global model against all simulated tracers, and thereby examine the potential impact of a "calibration" bias on various time scales, we neglect the data sparsity in our optimisation, and assume that the observations are - at least to some extent - representative of the average state of a larger geographic domain. By applying various metrics to a posteriori model assessment we evaluate the consequences of this assumption as detailed in the next section.

## 2.3 The misfit function and other metrics

Model misfit (i.e, the cost function applied during optimisation) was calculated by the root-mean-square error (RMSE) between simulated and observed annual mean tracers, mapped onto the respective three-dimensional model geometry (see also Kriest et al., 2017; Kriest, 2017; Kriest et al., 2020). By considering annual means we neglect any mismatch between the temporal variation of simulated and observed tracers; this choice was motivated by the climatological forcing of the applied circulation, which could not reproduce the temporal variability inherent in many of the observational data sets. Deviations between model and observations were weighted by the volume of each individual grid box, $V_i$, expressed as the fraction of total ocean volume $V_j^{\mathrm{T}}$ where observations exist for tracer $j$. The sum of weighted deviations was then normalised by the global mean concentration of the observational data of tracer $j$ ($\overline{o}_j$), and the resulting dimensionless numbers were added to provide the scalar misfit over all seven tracers, $J_{\mathrm{RMSE}}^{\mathrm{opt}}$:

$$J_{\mathrm{RMSE}}^{\mathrm{opt}} = \sum_{j=1}^{7} J_{\mathrm{RMSE}}^{\mathrm{opt}}(j) = \sum_{j=1}^{7} \frac{1}{\overline{o}_j} \sqrt{\sum_{i=1}^{N_j} (m_{i,j} - o_{i,j})^2 \frac{V_i}{V_j^{\mathrm{T}}}} \tag{1}$$

$j = 1, 2, ..., 7$ indicates the tracer type (phosphate, nitrate, oxygen, phytoplankton, zooplankton, dissolved and particulate organic phosphorus) and $i = 1, ..., N_j$ denotes the model grid boxes where observations exist. $o_{i,j}$ are the observations (in units of mmol P m$^{-3}$ for all tracers except nitrate and oxygen), and $m_{i,j}$ the model equivalents. For a model that deviates from the observations by less than the global mean value of each observed variable, the cost function value is smaller than 7.

In the different optimisations we considered different combinations of data sets and spatial domains for the evaluation of the misfit function $J_{\mathrm{RMSE}}^{\mathrm{opt}}$ (see below, section 2.4), yielding different properties of that metric. Therefore, the results obtained with Equation 1 for the different optimisation setups cannot be compared directly. For an independent and informative comparison of the optimal model performance, and to examine the consequences and limitations of the metric applied through Equation 1, we evaluated several other metrics a posteriori for the optimal model solutions:

Firstly, we calculated equation 1 for nutrients, oxygen, zooplankton and DOP in the upper 100 m, phytoplankton in the surface layer and POP through the entire vertical domain, regardless of whether these data were all considered during optimisation. We name this metric $J_{\mathrm{RMSE}}^{\mathrm{post}}$.

Secondly, we calculated several performance statistics such as the volume-weighted RMSE, the global bias $B$, the unbiased RMSE (RMSE', i.e. RMSE with bias subtracted), the normalised standard deviation $\sigma^*$ and the Pearson correlation coefficient $r$. These quantities are related through

$$\text{RMSE}'^2 = \text{RMSE}^2 - B^2 \qquad \text{and} \qquad \frac{\text{RMSE}'}{\sigma_\text{o}} = \sqrt{1 + \sigma^{*2} - 2\sigma^* r}$$

(Taylor, 2001; Jolliff, 2009), where $\sigma^* = \sigma_\text{m}/\sigma_\text{o}$ is the (spatial) standard deviation of the model data divided by that of the observations, and the global bias $B$ is given by the difference between volume-weighted average concentrations in the model and the observations. To compare the bias of different model components we alse evaluated the normalised bias $B^*$, i.e. the bias divided by the global mean of observations. Volume-weighting was obtained through Ferret's tools for averaging, calculation of variance and regression. We note that in the absence of any model bias RMSE will equal the unbiased RMSE', and the mismatch between model and observations indicates a mismatch between the amplitude (as given by the standard deviation) and the phase or spatial match (as given by the correlation coefficient) of two spatial patterns. Following Jolliff (2009), we therefore refer to $r$, $\sigma^*$ and RMSE' as "pattern statistics" or "pattern matching".

## 2.4 Experimental setup

Using MOPS coupled to ECCO TMs, Kriest et al. (2020) optimised six biogeochemical model parameters against global nutrients and oxygen, following a model spin-up of 3000 years (ECCO* in Kriest et al., 2020). To explore the effects of data sets and spin-up length we modified the setup by Kriest et al. (2020) and performed five optimisations that have the following common characteristics:

- Starting from observed inorganic tracer distributions (Garcia et al., 2006a, b) and globally constant organic tracer concentrations of $10^{-4}$ mmol P m$^{-3}$ (as in Kriest et al., 2020), model spin-up time was only 10 years before the evaluation of Equation 1.

- Because of the short spin-up time, and our focus on the adjustment of parameters related to surface processes, calculations of the models' fits to nutrient and oxygen concentrations were restricted to the upper 100 m.

These five optimisations differed with respect to i) their combinations of organic tracers considered in the misfit function, and ii) the number, type, and variational ranges of parameters to be optimised, Table 1. The five short-term optimisations are complemented by a sixth one, in which we spun up the model over 3000 years before evaluating an extended misfit function that includes nutrients and oxygen throughout the entire vertical domain, in addition to surface plankton and particulate organic matter (Table 1). As noted above, 3000 years of spin-up may even be too short to reach steady state. Indeed, a perfect optimisation setup would derive the spin-up time depending on, e.g., a Euclidean norm; however, this so far does not seem feasible in the current technical set-up of optimisation, where 10 model simulations with different sets of parameters run in parallel, and where the setup that is slowest to adjust would determine the computational demand of every iteration, resulting in

a large potential computational overhead. However, after 3000 years local tracer concentrations will only show small changes over time and with small effects on the misfit function (Kriest, 2017), even if the global inventories of oxygen and nitrate may still show some drift (Kriest and Oschlies, 2015). The six optimisations are identified by the length of the spin-up time ("S" for short and "L" for long), the number of parameters to be optimised (four or six), and – starting from optimisations against all tracer types ("All") the successive removal of specific tracer types ("-DOP", "-Org") or spatial domains ("SO", see below) from the misfit function:

**S6*-All: Wide boundaries for the particle flux exponent** In our initial optimisation we allow the detritus sinking speed, expressed through the particle flux exponent $b$ (see also Kriest and Oschlies, 2008, 2015), to vary between 0.5 to 1.8. The other five parameters subject to optimisation are the light affinity of phytoplankton (expressed as a half-saturation irradiance $I_c$), the maximum zooplankton growth rate ($\mu_{ZOO}$), two parameters that regulate production and decay of DOP ($\sigma_{DOP}$ and $\lambda_{DOP}$, respectively), and the oxygen demand for every mole phosphorus remineralised, $R_{-O2:P}$.

**S6-All: Narrow boundaries for the particle flux exponent** This setup is similar to S6*-All, but with values of the particle flux exponent $b$ being restricted to a lower bound of 1. With such restriction we follow findings of earlier optimisations where estimates of optimal $b$ ranged between 1 (Kwon and Primeau, 2006) up to 1.46 (Kriest et al., 2020).

**S6-DOP: Excluding DOP data** Based on S6-All we here exclude DOP data from Equation 1, which allows us to investigate the relevance of DOP data for constraining the estimates of the six parameters of interest. In particular, we are interested whether the two DOP parameters $\sigma_{DOP}$ and $\lambda_{DOP}$ can be efficiently constrained by a misfit function that lacks information on DOP.

**S4-SO: Excluding Southern Ocean phytoplankton data** For experiment S4-SO we reduce the number of parameters for optimisation, simply by adopting the best estimates of the DOP parameters $\sigma_{DOP}$ and $\lambda_{DOP}$ obtained from S6-All. Furthermore, we exclude DOP and chlorophyll data south of $40°$S, which embraces large HNLC (high nutrients low chlorophyll) regions where phytoplankton growth is limited by iron. With this approach we want to analyse whether the neglect of iron limitation in MOPS yields a bias in parameter estimates of the light affinity and/or zooplankton grazing. Such biased estimates may result from parameter adjustments that compensate for the unresolved iron limitation in HNLC regions. We note that in an intermediate step we carried out an optimisation similar to S4-SO, but with the Southern Ocean data included. This optimisation resulted in parameters which were very similar to those obtained from S4-SO, and hence produced very similar model results. We therefore do not include these in our further model analysis, but present the results of this complementary optimisation (here named S4-DOP) in Table S1 in the supplement.

**S4-Org: Excluding all organic data** Optimisation S4-Org tests the impact of organic tracer data and how they can affect optimal parameter estimates and model performance in general. The setup of S4-Org is similar to S4-SO, but it excludes observations of organic tracers everywhere, which leaves only surface data (0-100 m) of nutrients and oxygen to enter Equation 1.

**L4-SO: Including deep nutrients and oxygen** We finally investigate the impact of deep nutrient and oxygen concentrations on the optimal model solution by extending the model spin-up time to 3000 years and then consider global nutrients and oxygen in the evaluation of the misfit function. Otherwise the setup is the same as for S4-SO, i.e. with DOP and phytoplankton data excluded south of 40°S.

The six models setups with the respective optimised parameter sets were analysed at two time slices, namely after 10 and 3000 years of spin-up. With these cross-validation model runs we investigate the impact of spin-up time on the model performance on various time scales. In particular, we extend all model runs of the optimal S-scenarios to 3000 years; likewise, we also analyse model performance of L4-SO after 10 years of simulation. For a consistent comparison of model performance after optimisation we always include all tracers in the different metrics, regardless of the contribution to the misfit during optimisation.

**Table 1.** Experimental setup of different model runs and optimisations: range of parameters and data sets included in equation 1. Parameter values in italics are fixed, i.e. not subject to optimisation.

| | S6*-All | S6-All | S6-DOP | S4-SO | S4-Org | L4-SO | Unit |
|---|---|---|---|---|---|---|---|
| *Spin-up time* | | | | | | | |
| | | | 10 | | | 3000 | years |
| *Parameters* | | | | | | | |
| $I_c$ | | | 4-48 | | | | W m$^{-2}$ d$^{-1}$ |
| $\mu_{ZOO}$ | | | 1-3 | | | | d$^{-1}$ |
| $R_{-O2:P}$ | | | 150-200 | | | | mol O$_2$:mol P |
| $b$ | 0.5-1.8 | | | 1-1.8 | | | |
| $\sigma_{DOP}$ | | 0-0.5 | | | fixed at 0 | | |
| $\lambda_{DOP}$ | | 0.036-36 | | | fixed at 0.1848 | | y$^{-1}$ |
| *Observations* | | | | | | | |
| PO$_4$ | | | 0-100m | | full depth | | (mmol P) m$^{-3}$ |
| NO$_3$ | | | 0-100m | | full depth | | (mmol N) m$^{-3}$ |
| O$_2$ | | | 0-100m | | full depth | | (mmol O$_2$) m$^{-3}$ |
| DOP | 0-100m | | | - | | | (mmol P) m$^{-3}$ |
| surface Phy | | global | | 40°S-80°N | - | 40°S-80°N | mmol P m$^{-3}$ |
| Zoo | | | 0-100m | | - | 0-100m | (mmol P) m$^{-3}$ |
| POP | | | full | | - | full depth | (mmol P) m$^{-3}$ |

Optimisations were carried out using an evolutionary Estimation of Distribution Algorithm, namely the Covariance Matrix Adaption Evolution Strategy (CMA-ES; Hansen and Ostermeier, 2001; Hansen, 2006). A detailed description of how this algorithm is embedded in the coupled biogeochemistry-TMM framework is given in Kriest et al. (2017). Briefly, during each iteration ("generation") the algorithm defines a population of 10 individuals (10 biogeochemical parameter vectors of length

$n$ for $n$ parameters to be optimised), sampled from a multivariate normal-distribution in $\mathbb{R}^n$. For every individual parameter vector the model is run for a ten years period (in the S-scenarios) or 3000 years (L4-SO). Results of the corresponding model solution are evaluated via the misfit function (Equation 1). The statistical properties of the current population, as well as of previous generations, are used to update a mean vector and a (scaled) covariance matrix, whose elements reveal how sensitive the misfit function is to specific variations of parameter values. An ensemble of new parameter values is sampled according to the updated mean estimates and covariance matrix in $\mathbb{R}^n$ respectively. With the repeated updates, the parameter sampling space is gradually adapted towards a region of lower misfit function values until no further reduction of the lowest misfit function value can be achieved. Ideally, the mean of the final ensemble of parameter estimates represent the global minimum of the parameter-misfit function manifold.

## 3 Results and discussion

We first evaluate the performance of the optimisation procedure and compare the optimised model solutions against the best solution obtained with ECCO* in Kriest et al. (2020), for which global data of nutrients and oxygen were used and a model spin-up of 3000 years was considered. We then evaluate the contribution of the different data types to the model misfit function (Equation 1), as well as to the model performance measured by independent diagnostics: i) Pearson correlation coefficient, ii) unweighted RMSE against all observations, iii) biogeochemical fluxes, such as primary production or particle flux, iv) global oxygen inventory and v) OMZ volume. Results of these skill metrics are also contrasted with those obtained in earlier global model studies. We finally compare the spread of model solutions that arise from the distinct optimisation setups (Table 1) with those obtained on different simulation timescales.

### 3.1 Optimal parameter estimates and optimisation performance

The different optimisation setups generate parameter estimates that are all distinct from those originally derived for the reference ECCO* configuration (see Table 2). Optimal estimates of the half-saturation irradiance, $I_c$, are at least twice as high as that of ECCO*. These higher estimates reduce the light affinity of the phytoplankton in our ensemble of optimal model solutions. Apart from the S4-Org setup, optimal values of the maximum grazing rate $\mu_{ZOO}$ exceed the ECCO* estimate. A reduced light affinity together with an enhanced grazing pressure can be expected to reduce maxima in phytoplankton biomass. Only the S4-Org setup, for which all organic data are excluded from the misfit function, comprises a value for the maximum grazing rate that is much lower than that of ECCO*. Notably, it is this particular setup that also yields the highest possible value of the exponent of the particle flux profile ($b$=1.8), which enhances shallow remineralisation of particulate organic matter and reduces the particle flux to the ocean interior. For all other setups, estimates of $b$ remain lower (0.8 to $\approx$ 1) than the ECCO* reference value ($b$=1.46). Optimal estimates of $R_{-O2:P}$, which regulates oxygen demand of remineralisation, are always larger than that of ECCO*, with values for S6-DOP and S4-Org at its upper limit of 200 mol $O_2$:mol P. In general, best model fits to observations are achieved when the fraction of DOP production ($\sigma_{DOP}$), which regulates the production of DOP through zooplankton sloppy feeding and phytoplankton exudation, has values that are one third that of ECCO* or lower. We note that

**Table 2.** Results of different model optimisations: number of generations $L$ until optimisation convergence, final optimal metrics $J_{\mathrm{RMSE}}^{\mathrm{post}}$ (see Section 2.3) and optimal parameters (in italics: parameter values fixed in the respective optimisation). Values in squared parentheses show the range of parameter values for which the misfit function is within 0.1% of its minimum value. Note that the misfit $J_{\mathrm{RMSE}}^{\mathrm{post}}$ presented here includes every tracer and region, even if disregarded during optimisation. For comparison we also provide the parameter estimates and the misfit of the configuration ECCO$^*$ described in Kriest et al. (2020), which resulted from an optimisation against global nutrients and oxygen after a spin-up of 3000 years.

| | S6$^*$-All | S6-All | S6-DOP | S4-SO | S4-Org | L4-SO | ECCO$^*$ |
|---|---|---|---|---|---|---|---|
| $L$ | 140 | 138 | 349 | 87 | 115 | 39 | - |
| $J_{\mathrm{RMSE}}^{\mathrm{post}}$ | 6.030 | 6.052 | 6.082 | 6.046 | 6.874 | 6.125 | 7.072 |
| Optimal parameters: | | | | | | | |
| $I_{\mathrm{c}}$ | 34.44 | 33.98 | 31.57 | 28.84 | 38.38 | 22.11 | *9.65* |
| | [ 31.4- 37.4] | [ 31.5- 35.0] | [ 28.3- 34.7] | [ 26.0- 31.8] | [ 32.4- 39.9] | [ 17.2- 25.2] | |
| $\mu_{\mathrm{ZOO}}$ | 2.801 | 2.594 | 2.895 | 2.807 | 1.021 | 2.369 | *1.893* |
| | [ 2.63- 2.87] | [ 2.45- 2.72] | [ 2.63- 3.00] | [ 2.44- 3.00] | [ 1.00- 1.33] | [ 2.10- 2.79] | |
| $R_{-\mathrm{O2:P}}$ | 187.8 | 188.3 | 200.0 | 200.0 | 189.5 | 169.3 | 151.1 |
| | [ 185.9- 193.6] | [ 181.3- 190.4] | [ 194.0- 200.0] | [ 184.9- 200.0] | [ 180.0- 190.6] | [ 161.8- 175.9] | |
| $b$ | 0.803 | 1.000 | 1.000 | 1.000 | 1.800 | 1.024 | 1.461 |
| | [ 0.77- 0.83] | [ 1.00- 1.02] | [ 1.00- 1.04] | [ 1.00- 1.03] | [ 1.78- 1.80] | [ 1.00- 1.09] | |
| $\sigma_{\mathrm{DOP}}$ | 0.028 | 0.000 | 0.049 | *0.000* | *0.000* | *0.000* | *0.150* |
| | [ 0.02- 0.07] | [ 0.00- 0.00] | [ 0.00- 0.07] | | | | |
| $\lambda_{\mathrm{DOP}}$ | 0.238 | 0.184 | 0.168 | *0.184* | *0.184* | *0.184* | *0.170* |
| | [ 0.20- 0.34] | [ 0.15- 0.22] | [ 0.12- 0.24] | | | | |

another source of DOP is a linear mortality rate of phyto- and zooplankton of $0.01\,\mathrm{d}^{-1}$ (see Kriest and Oschlies, 2015). Values
of the DOP decay rate ($\lambda_{\mathrm{DOP}}$) are almost indifferent from ECCO$^*$, except for a higher rate estimate in the S6$^*$-All setup.

The efficiency of the optimisation procedure can be deduced from the number of generations required for convergence ($L$ in Table 2). The number of generations for identifying an optimal solution is not affected by the imposed range of possible parameter values for $b$, as apparent when comparing setup S6-All ($L$=138) and the setup S6$^*$-All ($L$=140). Clearly, DOP data are helpful to constrain the full set of six model parameters to be optimised. This is indicated by the large number of iterations
($L = 349$) needed for the optimisation S6-DOP. Yet, even in the absence of DOP data and despite a larger number of iterations, our results reveal that rate estimates of DOP production and decay can be obtained, similar to estimates achieved with DOP data included. When fixing values of the two DOP parameters to their best estimates of S6-All, the number of parameters to be optimised is reduced to four (S4-SO and S4-Org). Because of the reduced dimensionality of the problem convergence is now accomplished faster ($L$=87 and 115, respectively). The fastest convergence is achieved by extending the model spin-up time
and by the additional consideration of the deep nutrients in L4-SO. However, this comes at the cost of a 300-fold increase in simulation time.

Regardless of the optimisation setup, values of the minima of the normalised misfit function $J_{\mathrm{RMSE}}^{\mathrm{post}}$ are mostly similar among the different model setups optimised against organic tracers, when the misfit is evaluated a posteriori for all tracers and regions (Table 2 and Figure 1). Compared to solution of ECCO$^*$, which shows a misfit of 7.072, $J_{\mathrm{RMSE}}^{\mathrm{post}}$ decreases between 315   14% (S6-DOP, $J_{\mathrm{RMSE}}^{\mathrm{post}} = 6.082$) to 15% (S$^*$6-All, $J_{\mathrm{RMSE}}^{\mathrm{post}} = 6.03$) in optimisations that include organic tracers (Table 2 and Figure 1). Excluding organic tracers in the optimisation (S4-Org) results in a misfit that is only a few percent smaller than that of ECCO$^*$, an optimised solution that disregarded organic tracers as well. Thus, a 14% to 15% reduction of the misfit appears as a robust result, when introducing organic tracers as additional constraints.

To summarise, considering the trade-off between simulation time (S- vs. L-setups), number of iterations required ($L$ in 320   Table 2), and improvement of the misfit (Equation 1) for all tracers and regions, apparently a good prior estimate of the production and decay parameters for DOP as in S4-SO helps to achieve a reasonable model fit to observations, while keeping the computational costs relatively low. In contrast, the omission of organic observations from the misfit function, as in S4-Org, deteriorates the model fit to observations in the upper 100 m and on shorter time scales.

## 3.2   Optimal model performance for various surface metrics on different time scales

Improved fits to surface nutrients are achieved from the optimisations, exhibiting a substantial reduction in $J_{\mathrm{RMSE}}^{\mathrm{post}}$ down to about one half that of ECCO$^*$ (Figure 1 and Table S2). The improvement in representing plankton concentrations is less pronounced (less than 10%), in contrast to a reduction of $J_{\mathrm{RMSE}}^{\mathrm{post}}$ of DOP by 25% (Table S2). Much of the adjustment is due to a reduction in normalised bias $B^*$ (see section 2.3), which decreases to less than 5% that of ECCO$^*$ for plankton and DOP, and becomes less than 20% for nutrients (Figure 1 and Table S2). However, the inorganic tracers' contribution to the total misfit 330   function $J_{\mathrm{RMSE}}^{\mathrm{post}}$ is small, whereas the organic tracers – in particular: POP and zooplankton – still have a considerable residual misfit.

The large remaining misfit can be explained by a lack of spatial correlation between observed and modelled patterns, as evident from Taylor diagrams in Figure 2. Even after optimisation the correlation coefficients $r$ (see section 2.3) of the different organic components remain low, between 0.3 and 0.4 for phytoplankton and less than 0.2 for zooplankton. Likewise, RMSE' 335   (RMSE with bias subtracted) is quite large. Thus, an optimisation, with observations of organic tracers included, does not reduce the pattern error ($r$ and RMSE'), which explains the similar performance of S4-Org compared to the other model setups.

Simulated organic tracers exhibit a spatial variability $\sigma^*$ (see Section 2.3) that is much lower compared to observations in all setups where organic tracers are considered (Figure 2). The solution of S4-Org yields a greater spatial variability of 340   (phyto)plankton and POP, which actually agrees better with the observed variability. The larger variability of organic components in S4-Org likely arises because of its large optimal value for $b$, which triggers shallow remineralisation, and thus a larger nutrient supply to the surface. Combined with a low grazing rate this allows phytoplankton to reach higher phytoplankton biomass (Figure S1), thereby generating larger variance.

All metrics ($J_{\mathrm{RMSE}}^{\mathrm{post}}$, RMSE', correlation $r$, normalised standard deviation $\sigma^*$ and normalised bias $B^*$) show similar re- 345   sponses to parameter changes after a simulation time of 3000 years (Figures 1 and 2, right panels). Therefore, a spin-up length

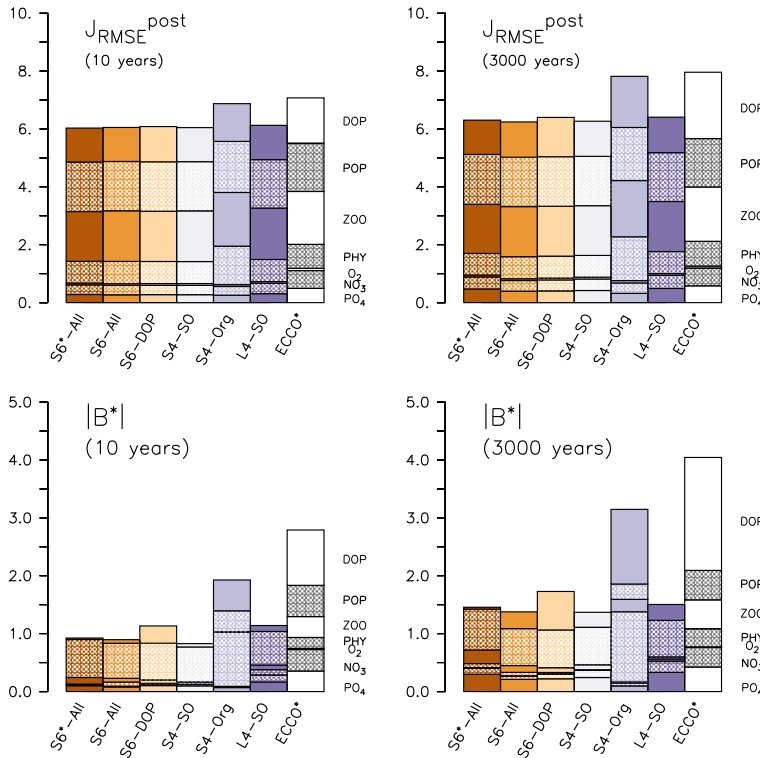

**Figure 1.** Components of $J_{\mathrm{RMSE}}^{\mathrm{post}}$ (top) and normalised bias $B^*$ (bottom) of the optimised model runs, and from ECCO$^*$ by Kriest et al. (2020). Left: after 10 years, right: after 3000 years. The different fractions of the bars denote the contribution of each tracer $j$ to the misfit. Note that metrics have been evaluated for every tracer and region, even if these where not considered in the optimisation (DOP in S6-DOP, S4-SO, S4-Org and L4-SO; organic tracers in S4-Org; phytoplankton south of $40°$S in S4-SO and L4-SO).

of 10 years seems to be sufficient to examine model skill with respect to surface metrics. However this stability can be due to the low sensitivity of the metrics because of the coarse circulation, that may dampen the differences at various time scales.

Compared to S4-SO, optimisation L4-SO combined two changes to the misfit function, namely a change of spin-up length before evaluation of the misfit function and the consideration of deep inorganic tracers. The combination of these two changes
was necessitated by the long time scales of deep ocean processes and circulation, which require a long spin-up to induce effects on deep tracers. To disentangle the effects of these changes, one could (i) either consider deep inorganic tracers after a short spin-up time, or (ii) spin-up the model over 3000 years and apply the same metric as in S4-SO (surface nutrients and organic tracers). For the first case (i) we note that after short spin-up time simulated deep tracers would still be very near the model's initial condition, resulting in a misfit that is less informative, and not very sensitive to changes in model parameters.
Considering the alternative case (ii), an extended spin-up time applied with the same metric as in S4-SO may likely result in surface metrics similar to those obtained with a short spin-up time. This is indicated by Figures 1 and 2, which show posterior surface metrics of L4-SO that are quite similar on both timescales. In addition, they are also quite similar to those of S4-SO,

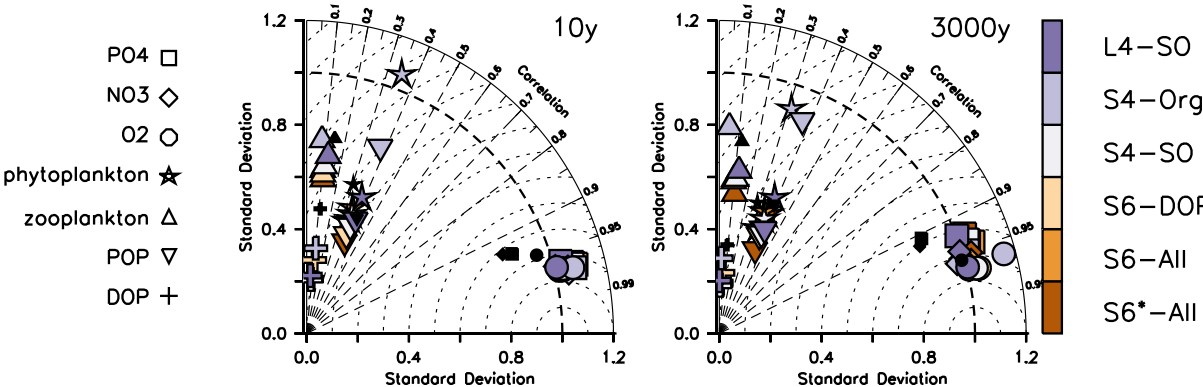

**Figure 2.** Taylor diagrams for annual mean oxygen (circles), phosphate (squares), nitrate (diamonds), phytoplankton (stars), zooplankton (triangles), DOP (pluses) and POP (inverted triangles) after 10 years (left panel) and after 3000 years (right panel). Regardless of optimisation setup (S- vs. L-setups), the metrics shown in this plot consider only the fit in the upper 100 m, except for that of POP, which always covers the entire vertical domain, and phytoplankton, which is based on phytoplankton in the first model layer (0-10 m; see Table 1). Colours indicate model solution from the optimisations listed in Table 1. Small black symbols additionally show results of ECCO*. Normalised standard deviation $\sigma^*$ is displayed on the x- and y-axis. The correlation coefficient $r$ is indicated by as radial dashed lines (azimutal position), and the (unbiased) RMSE' is visible as the distance of a symbol from "1" on the abscissa. Note that metrics have been evaluated for every tracer and latitude, even if these where not considered in the optimisation, as explained in section 2.3.

which applied a shorter spin-up time. These results suggests that the consideration of deep nutrients and oxygen in L4-SO is especially important for the parameter regulating the respiratory oxygen demand $R_{-O2:P}$, but does not strongly affect the
biogeochemical turnover and tracer distribution at the surface.

Figure 3 indicates the best model setups with respect to the different metrics. In general, solutions from optimisations against organic tracer data perform well for many metrics regarding organic tracers. On the other hand, S4-Org, that considers only the misfit of dissolved inorganic tracers, performs best for the bias (expressed as $|B|$), $J_{\mathrm{RMSE}}^{\mathrm{post}}$, $r$ and RMSE' of nutrients. Surprisingly, L4-SO outperforms the other model setups with regard to the correlation between observed and simulated plankton and
POP (Figure 3), despite the fact that optimisation also has to consider deep nutrients and oxygen. A good correlation between observed and simulated POP and zooplankton is also obtained with ECCO*, which was not optimised against any organic tracers Kriest et al. (2020), but only against global inorganic tracers. This points towards a potential tight coupling between POP (and its sinking flux) and global nutrient and oxygen distribution, in agreement with earlier studies (Kwon and Primeau, 2006; Kriest et al., 2012). Yet, in many cases metrics that are closely related to spatial surface patterns, such as RMSE', $J_{\mathrm{RMSE}}^{\mathrm{post}}$ and
$r$, are quite insensitive to the optimisation strategy (see hatched patterns in Figure 3)The small difference in model outcomes with respect to the correlation coefficient $r$ of nutrients and oxygen can be explained by the fact that $r$ was already high prior to optimisation, leaving little room for further improvement of the overall spatial patterns (see also Figure 2). Thus, all model setups remain, more or less, at the same low-level model fit with regard to pattern-matching metrics, whereas global (integral) metrics such as the bias or the normalised standard deviation are more sensitive to the optimisation strategy.

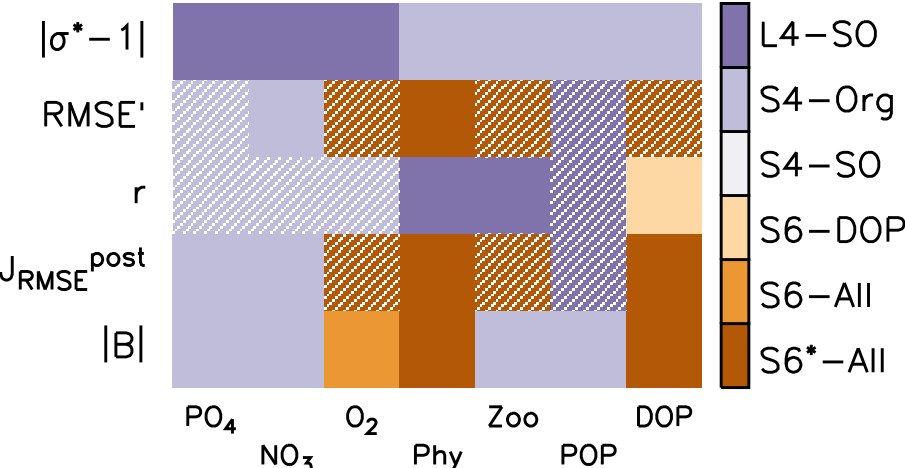

**Figure 3.** Summary of optimal model setups after 10 years of simulation for different metrics (y-axis) and tracers (x-axis). The former include the deviation of normalised model standard deviation from 1 (absolute values), RMSE' and Pearson correlation coefficient $r$ (see also Figure 2), normalised RMSE $J_{\text{RMSE}}^{\text{post}}$ and global bias $B$, expressed as absolute value (see also Section 2.3). The colour code indicates the best model solution for the respective tracer and metrics. Hatched fields indicate that, within that specific metric, the respective tracer range is less than 10% of the average of all six model setups.

The metrics for surface nutrients obtained in our experiments agree well with those of other global model studies (see Table S3), in particular with regard to the high correlation coefficient (between 0.93-0.96), small bias and a normalised variance around 1. Global model studies also sometimes report model performance with respect to chlorophyll, but not always in the same way. Some studies report a correlation of log-transformed chlorophyll between 0.6 and 0.7 (Dunne et al., 2013; Moore et al., 2013; Aumont et al., 2015), which is higher than we could achieve with our optimisations ($r$ of log-transformed phytoplankton between 0.36 to 0.48 after 10 years of simulation, and between 0.27 and 0.51 after 3000 years). Correlation of untransformed chlorophyll varies much more, from high values of $\approx 0.85$ (Le Quere et al., 2016, 10 year average after a model spin-up of 10 years) to lower values around 0.3 to 0.4 (Yool et al., 2013; Seferian et al., 2013). When analysing the performance of six different BGC models in a common circulation, Kwiatkowski et al. (2014) found variable fits to observations, resulting in a range of $r$ between almost zero to 0.5 (see also Table S3). With $r$ ranging between 0.3 and 0.4 our model results are at the lower end of this range. The quite low spatial variability observed in our study agrees with the low variability reported for half of the models tested by Kwiatkowski et al. (2014).

For organic tracers other than chlorophyll an evaluation of model skill has been carried out less often. Stock et al. (2014), Aumont et al. (2015) and Stock et al. (2020) report a fit of simulated to observed mesozooplankton with $r$ between $0.37 - \approx 0.5$, which is higher than the fit of our model ensemble (generally around 0.1). The models applied in these studies all distinguish between meso- and microzooplankton. This indicates that a more complex parameterisation of zooplankton, and thus a larger number of free parameters, may improve the model fit to corresponding data.

After fitting a global model against observed DOC, DON and DOP, Letscher et al. (2015) achieved correlations of log-transformed data of semi-labile DOP in the upper 500 m between 0.3 and 0.44, which is higher than in our study (0.18 to 0.22 and 0.15 to 0.20 for spin-up times of 10 and 3000 years, respectively). Depending on model setup, the bias in their model solutions ranged between -25% up to 136% of the observed value (Letscher et al., 2015, their Table 4). With regard to $r$, the performance of our optimised model solutions appears seemingly poor, but at the same time the bias could be reduced down to -3% and 3% for S6$^*$-All after 10 and 3000 years, respectively. For comparison, the bias in the ECCO$^*$ solution is still 96% (195%) after 10 (3000) years.

In summary, our model representations of surface nutrients and DOP benefit most from optimisation, yet much of the improvement especially for DOP is due to an extensive reduction of the bias. The pattern error, as expressed through RMSE' or $r$, improves slightly for the inorganic tracers, but hardly for the organic tracers. This lack of improvement in simulated patterns of the organic tracers can likely be attributed to the coarse spatial resolution ($1° \times 1°$) and climatological circulation applied in the experiments. In contrast to this, observations especially of zooplankton, DOP and POP depend on local hydrodynamics and episodic events which are not resolved by this type of model. This seems to impede an improvement towards observed patterns, leaving only room for an improvement in the bias.

A potential solution to this problem could be to apply optimisation to a coupled physical-biogeochemical model that more realistically resolves the physical environment at high spatial and temporal scales. However, such a model could hardly be simulated at a global scale over more than a few decades. In addition, even a model that resolves eddies and filaments well may not exactly reproduce the position and timing of mesoscale physical features such as eddies and filaments, again with consequences for biogeochemical variables, and hence the pattern-matching statistics such as RMSE' or correlation.

A misfit function that involves point-wise, local data-model residuals, such as the $J_{\mathrm{RMSE}}^{\mathrm{opt}}$ applied during optimisation, will attribute to tracers some high error caused by the dynamical representation of physics that does not resolve local real-time and (sub)mesoscale dynamics (representation error). Because of the large contribution of the pattern errors to the overall $J_{\mathrm{RMSE}}^{\mathrm{opt}}$, most parameter estimates are presumably biased. We speculate that the introduction of observational error information appears to be essential for overcoming the problem of paucity of the organic data, which is apparently less critical for the large number of global, objectively analysed, inorganic data of nutrient concentrations. Such observational error information may combine uncertainties in measurements (or in observational data products) with the representation error that accounts for spatio-temporal variability unresolved by the model. This would require switching from RMSE to a likelihood-based metric. Rather than summing up point-wise local residuals, combining data to describe their statistical properties on regional scale, is one way of obviating pattern errors to affect parameter estimation. The problem of pattern errors affecting parameter estimates can be reduced if means of, possibly log-transformed, data of specified ocean regions are combined with spatial variance information, e.g. as in Chien et al. (2022), who considered biomes derived by Fay and McKinley (2014) as regional entities.

### 3.3 Global biogeochemical fluxes

After a spin-up of 10 years all experiments except S4-Org exhibit lower global primary production than the observational estimates (Figure 4). The underestimation of primary production in high latitudes could be caused by the low light affinity of

phytoplankton (Table 2), which affects phytoplankton growth especially in high latitudes. This is less expressed in the tropics and subtropics where light limitation plays a smaller role. Interestingly, the setup with the lowest light affinity, S4-Org, reveals the highest global (Figure 4) and zonally averaged (Figure S2) primary production, indicating the reasons for the variations in this flux have to be sought elsewhere. Most likely, shallow remineralisation induced by $b = 1.8$ in S4-Org increases subsurface

nutrient concentrations and hence nutrient supply to the euphotic zone, thereby increasing production especially in the tropics (Figure S2). Extending the model runs to 3000 years further enhances global primary production in the solution of S4-Org. Here, the northward transport of unutilised nutrients from the Southern Ocean, caused by shallow remineralisation and light limitation, especially in S4-Org, might play a role (see Figures S3 and S2, that show zonally averaged inorganic tracers and biogeochemical fluxes); a feature that has already been noticed by, for example, Keller et al. (2016). Global primary production

of our model ensemble is somewhat at the lower end of the large range of 28 and 82 Pg C y$^{-1}$ obtained with other global models (see also Table S4), but within the range of more recent estimates between 22 to 57 Pg C y$^{-1}$ of CMIP6 models (Seferian et al., 2020).

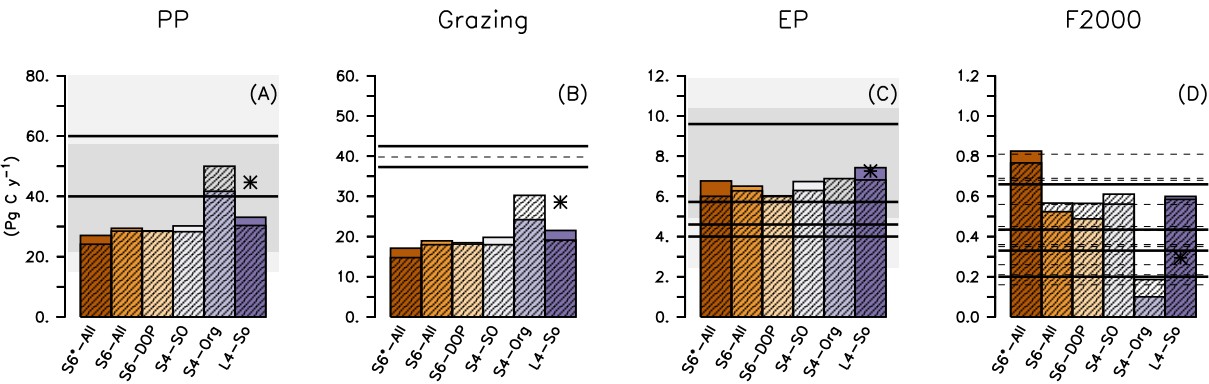

**Figure 4.** Global annual biogeochemical fluxes (Pg C y$^{-1}$) of primary production (A), zooplankton grazing (B), export production (C) and particle flux at 2000 m (D) of different model setups. Coloured bars show flux after 10 years (as in the optimisations) and hatched bars the flux when the model is simulated with the same parameters over 3000 years. Straight lines denote observational flux estimates by Carr et al. (2006, PP), Steinberg and Landry (2017, grazing by micro- and mesozooplankton), Lutz et al. (2007, EP), Dunne et al. (2007, EP and F2000), Honjo et al. (2008, EP and F2000), Henson et al. (2012, EP and F2000) and Guidi et al. (2015, F2000). Thin dashed lines show results of other global model studies listed in Table S4. Light and dark shaded areas in panels (A) and (C) indicate the range of fluxes simulated by CMIP6 and CMIP6 models, respectively (Seferian et al., 2020, their Table 4). Star indicates results of ECCO$^{*}$ by Kriest et al. (2020).

The high maximum grazing rates (Table 2) cannot compensate impacts of the low primary production on global grazing fluxes in model setups S6*-All to S4-SO and L4-SO, resulting in a global grazing flux that is only about half of the observed

estimate. We note that the grazing estimates by Steinberg and Landry (2017) are based on an assumed global primary production of 50 Pg C y$^{-1}$ (i.e., much larger than simulated by most of our model experiments); however, even when correcting our results of global grazing by the respective underestimates in production, the models would remain biased low. Also, our simulated global grazing turns out to be lower than the grazing simulated by Aumont et al. (2015) (see Figure 4 and Table S4) or

Stock et al. (2014). Both models by Aumont et al. (2015) and Stock et al. (2014) distinguish between two or three zooplankton classes, whereas in our study all zooplankton types are aggregated into a single class, representing large, motile organisms such as copepods, as well as microzooplankton such as ciliates and flagellates. Explicitly resolving the latter group with its higher grazing rates and turnover could result in a larger global grazing of zooplankton; for example, in the study by Aumont et al. (2015) microzooplankton contributes to more than 90% of total grazing on phytoplankton, and 78% in the study by Stock et al. (2014). Together with the underestimated zooplankton biomass noted above, this indicates that a more detailed representation of zooplankton might provide a more realistic potential link to models of higher trophic levels (as, e.g., in Stock et al., 2017).

In contrast to primary production and grazing, global export production is similar in all model simulations (Figure 4), despite the considerable differences in optimal $b$ (Table 2). The similarity in export production arises mainly from compensating effects of lower primary production but faster sinking, for the model runs with $b \lesssim 1$. For S4-Org (with $b$=1.8), the enhanced primary production is clearly compensated by the slower sinking of particles. Our results are thus consistent with the notion that for all reasonable parameter settings, it is the ocean circulation that controls new production and export production (Oschlies , 2001; Najjar et al., 2007; Kriest et al., 2020)

Global particle flux at 2000 m simulated by our different model setups depends strongly on the optimal value of $b$, with the largest flux obtained by $b = 0.8$ (S6*-All), and the lowest with $b = 1.8$ (S4-Org). The flux of the four model configurations with $b \approx 1$ falls within values suggested by Honjo et al. (2008, 0.43 PgC y$^{-1}$) and Henson et al. (2012, 0.66 PgC y$^{-1}$), but is larger than the estimates by Dunne et al. (2007, 0.2 PgC y$^{-1}$) and Guidi et al. (2015, 0.33 PgC y$^{-1}$). The low value of $b = 0.8$ in setup S6*-All causes a deep particle flux that exceeds the high global estimate by Henson et al. (2012), whereas S4-Org, with $b = 1.8$, simulates a global flux that is below all other estimates at 2000 m depth. Except for the extreme values of S4-Org and S6*-All, the range of simulated global particle flux coincides with that of other model studies, ranging between 0.16 and 0.81 Pg C y$^{-1}$ (see also Table S4).

The range in particle flux profiles applied in the different model setups also affects the (particle) transfer efficiency TE, as calculated by dividing the simulated particle flux at 1000 m by the particle flux at a depth 100 m. Dividing global mean particle flux at these two depths, our model simulations with $b = 1$ and after a spin-up of 10 years exhibit a TE of 0.17-0.18, which agrees with the observed range reported by Wilson et al. (2022). Considering the full set of our model experiments results in TE between 0.06 (S4-Org with $b = 1.8$) up to 0.23 (S6*-All, $b = 0.8$). This variation is almost as large as that of different global models analysed by Wilson et al. (2022), which vary between TE=0.03 (UKESM1-0-LL) and and TE=0.25 (IPSL-CM5A2-INCA).

We note that when diagnosing TE from simulated model fluxes a considerable fraction of the transfer of particulate organic matter is due to other processes than particle sinking. While theoretically it seems straightforward to derive the (nominal) TE directly from the applied particle flux profile (for example, $b = 1$ would result in TE=0.1), TE diagnosed from our simulated model fluxes exceeds this value by 70 to 80%. The discrepancy between the nominal and diagnosed TE arises from many facts, for example additional physical transport of particulate organic matter through mixing or upwelling, reduced remineralisation in oxygen minimum zones, as well as numerical diffusion (Kriest and Oschlies, 2011). Because of the very variable setup of global models, that differ in resolution, physical and biogeochemical aspects, it may be difficult to disentangle the reasons for

their divergence in TE. However, our results suggest that variations in model parameters (especially the particle flux parameter) may account for a considerable fraction, but not all, of the variation in TE.

Given the many assumptions that go into observed global estimates of deep particle flux, and the potentially strong effect of this flux for transfer efficiency and carbon storage, we additionally examined simulated particle flux against three different data sets of sediment trap observations (see section A2). While the correlation coefficient of ECCO* to observations ranged between 0.22 to 0.28 for the three different data sets (no figure), optimisation improves the correlation between simulated and observed particle flux to values between 0.3 and 0.41 after 10 years (Figure 5 and Figures S4, S5 and S6), which declines to 0.24 to 0.39 after 3000 years. These correlations are only slightly better than those found by Schwinger et al. (2016), who tested four different sinking parameterisations in a subjectively tuned global model, and found $r$ between 0.11 and 0.32 (the highest value was achieved with a spatially varying particle flux length scale). Depending on $b$, in agreement with the comparison for global flux at 2000 m (Figure 4), particle flux averaged over the individual sites and depths can be biased very high (S6*-All with $b = 0.8$) or low (S4-Org, $b = 1.8$), and is otherwise between 66 and 212 mmol C m$^{-2}$ y$^{-1}$ (Figures S4, S5 and S6). After 3000 years both bias and variance show a smaller range of the model ensemble (see Figures 4 and 5).

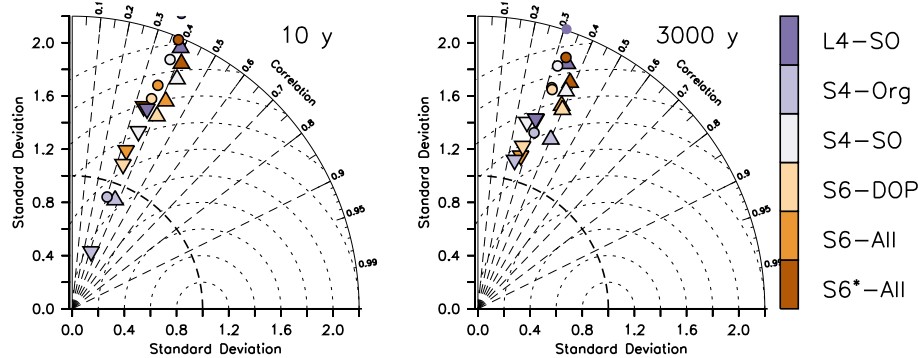

**Figure 5.** Taylor diagrams for simulated particle flux after a spin-up of 10 years (left panel) and 3000 years (right panel) compared to data set by Honjo et al. (2008, inverted triangles), Lutz et al. (2007, triangles) and Mouw et al. (2016, circles). Colours indicate model type.

Thus, for model setups with $b \approx 1$ or less, the turnover at the sea surface seems to be reduced and the transport to the ocean interior might be too high, at least when applying a particle flux exponent of $b = 0.8$. Given that differences in deep particle flux between the model setups are larger in year 10 than in year 3000, this diagnostic could serve as a quite robust estimator, that may constrain model solutions even after short spin-up periods. However, the spread of observed particle flux to the ocean interior is large, and the uncertainties in these observations remain high (e.g. Kähler and Bauerfeind, 2001; Scholten et al., 2001; Buesseler et al., 2008; Siegel et al., 2008); we thus so far lack a strong observational constraint on this global flux.

### 3.4 Oxygen inventory and OMZ volume

On long timescales, nutrient and oxygen concentrations in the deep ocean are mainly determined by the value assigned to the particle flux exponent $b$ and the large scale circulation (e.g. Kwon and Primeau, 2006; Kriest et al., 2012). Such model

behaviour is also evident from the normalised standard deviation of global nutrient distribution that varies between $\approx$0.8 to 1.2 for phosphate and nitrate, with S4-Org ($b = 1.8$) showing a strong underestimate of spatial variance (Figure 6). Despite these variations, all model setups show similarly high correlations to observations and low RMSE'. Considering oxygen, the models differ mostly with respect to the correlation coefficient and RMSE'; again, S4-Org shows the largest deviation to observations, whereas the other model setups are more or less similar. Even though S4-Org is the only optimisation setup that exclusively targets at inorganic tracers (but only for the surface), its solution shows the worst match to observations at a global scale and after 3000 years of model spin-up. The best model performance with regard to global nutrients is obtained with either S6-All, S6-DOP or S4-SO (i.e., model setups with $b = 1$), but L4-SO clearly outperforms all other setups with regard to the global oxygen distribution and variance.

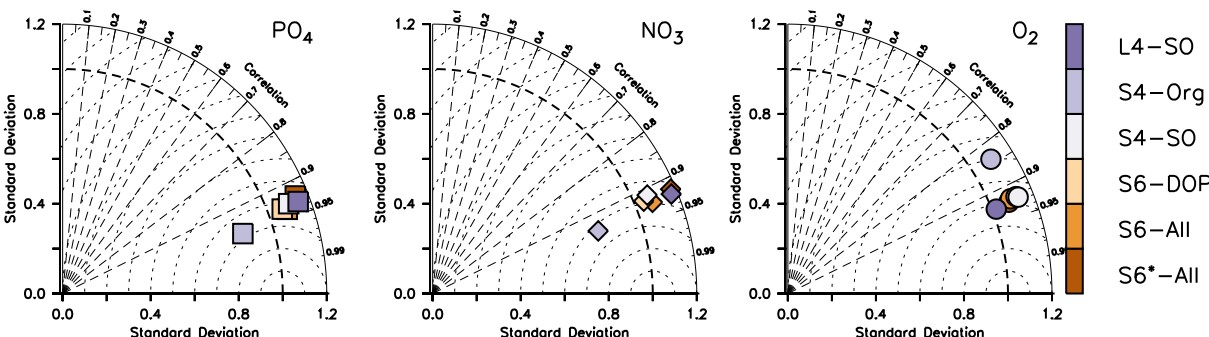

**Figure 6.** Taylor diagrams for annual mean phosphate (left), nitrate (middle) and oxygen (right) after 3000 years, and analysed over the global domain. Colours indicate model type. Note that metrics have been evaluated for every tracer and latitude, even if these where not considered in the optimisation.

In addition to tracer distributions, the global nitrate and oxygen inventory may also be affected by changes in model parameters (Kriest and Oschlies, 2015). Indeed, the six optimal model setups presented here show significant differences in globally averaged oxygen concentrations: after 3000 years, the oxygen bias varies between -13.6 and 24.1 mmol $O_2 m^{-3}$ (Figure 7), i.e. between about -8% to 14% of the global observational mean. We emphasize that this variation arises solely from changes in biogeochemical model parameters, and are not a result of circulation changes. Again, results of L4-SO out-compete the results of all other model setups with regard to this metric. The solution of L4-SO includes a good representation of nitrate (Figure S7), and the good match of the oxygen does not come at the cost of nitrate as complementary oxidant. Setup S4-Org, which targets only at surface inorganic tracers, performs worst with respect to nitrate (Figure S7). Thus, to achieve a good fit for model estimates of the global oxygen and/or nitrate inventory after millennial simulation times it seems necessary to either consider the global data and long spin-up times for calibration (as in L4-SO), or – in case of short spin-up times – also include organic tracer data (as in S6*-All, S6-ALL, S6-DOP or S4-SO).

The bias range found in our study is similar to that of many models analysed by Bopp et al. (2013), though in their study one model exhibited a very large positive bias of more than 53 mmol m$^{-3}$, and two models were biased very low, down to $-42$ mmol m$^{-3}$, leading to an overall model spread of 96 mmol m$^{-3}$. We note that in our study changes in oxygen inventory

are only induced by biogeochemical parameter changes, whereas Bopp et al. (2013) report values for models of different
complexity, circulation, forcing and spin-up time, which explains the larger variation of oxygen in that study.

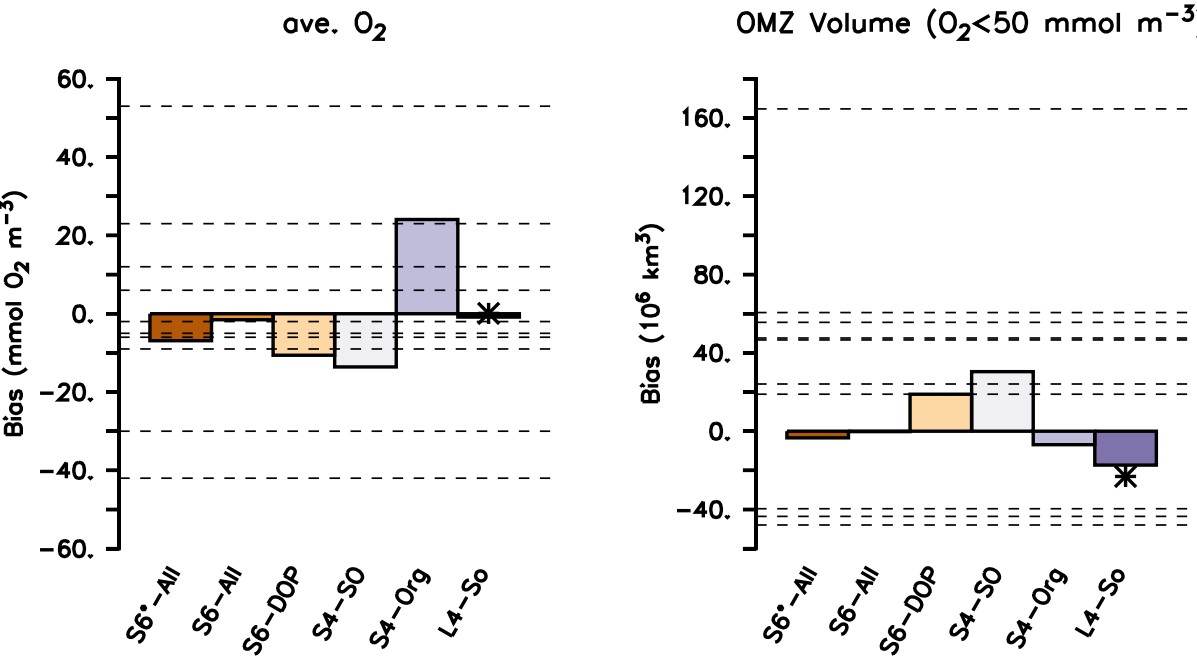

**Figure 7.** Oxygen bias (left) and bias of OMZ volume, with OMZs defined $O_2 < 50\,\mathrm{mmol\,m^{-3}}$ (right). Global bias $B$ and OMZ volume calculated from the difference between simulated model results after a spin-up of 3000 years and the observed values from Garcia et al. (2006b). A star indicates the performance of ECCO* by Kriest et al. (2020). Horizontal dashed lines indicate values listed for the different models in Table 2 of Bopp et al. (2013).

Differences in global oxygen bias between the model setups (Figure 7) do not mirror those for primary production, export production, grazing or particle flux (Figure 4). This finding reinforces the complex and nonlinear nature of processes that determine the ocean's oxygen concentration, even under climatological forcing. Indeed, over the time scale from 10 to 3000 years the trajectories over time of global oxygen bias differ substantially among the different model setups (Figure 8), and can be roughly sorted into three groups. The first group, consisting of S4-SO, S6-DOP and S6*-All, shows averaged oxygen concentrations that decrease with time. These three models are characterised by either a high oxygen demand of remineralisation ($R_{-O2:P} = 200$ mol $O_2$:mol P) together with a particle flux exponent of $b = 1$ or a moderately high $R_{-O2:P}$ of 187.8 mol $O_2$:mol P and $b = 0.8$ (S6*-All). In the second group (L4-SO and S6-All, both with $b \approx 1$ and a moderate $R_{-O2:P}$) the oxygen bias first decreases until about year 2000 and then increases afterwards, until approaching almost zero drift in year 3000. In the remaining third group, which only consists of S4-Org, oxygen is increasing over the entire trajectory, likely owing to the very shallow remineralisation depth induced by $b = 1.8$, in conjunction with a moderate value for $R_{-O2:P}$ of 189.5 mol $O_2$ : mol P. Focusing on individual model trajectories, the oxygen bias varies over time between 4.4 mmol $O_2$ m$^{-3}$

(L4-SO) to 24.1 mmol $O_2$ m$^{-3}$ (S4-Org), or between 3 to 14% of the observed value; the variation decreases to less than 5% if we restrict our analysis to the shortest spin-up time applied in CMIP6 of 150 years (see large vertical bars at the abscissa in Figure 8). After 3000 years the model solutions have diverged by 37.7 mmol $O_2$m$^{-3}$ or 22% of the observed global average oxygen concentration. We note that at this time some setups still show considerable trends. To summarise, in our study maximum variations caused by spin-up length (24.1 mmol $O_2$m$^{-3}$) and model setup (37.7 mmol $O_2$m$^{-3}$ at year 3000) are considerable, and could potentially explain 25% and 40% of the spread observed by Bopp et al. (2013).

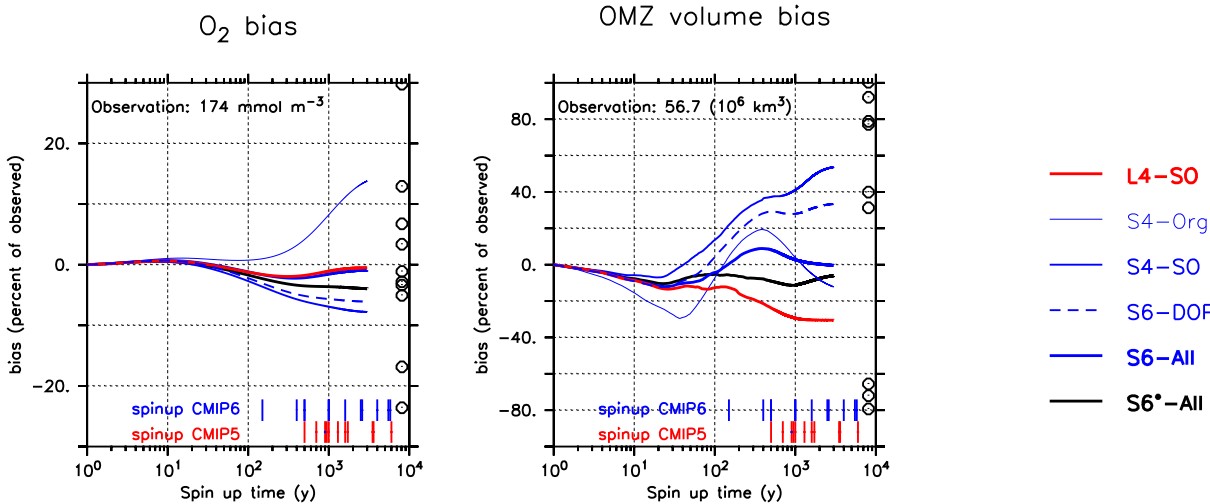

**Figure 8.** Oxygen bias (left panel) and OMZ volume bias (defined by $O_2 < 50$ mmol $O_2$m$^{-3}$; right panel) over the entire model trajectory of 3000 years (in log scale). Bias calculated with reference to Garcia et al. (2006b). Circles indicate values listed in Table 2 of Bopp et al. (2013). Vertical red and blue bars at the abscissa denote the model spin-up times of different CMIP5 and CMIP6 models listed in Seferian et al. (2020). Numbers give the observed value from Garcia et al. (2006b, i.e., the reference data set used in this study).

The global OMZ volume bias is more sensitive to parameter changes and spin-up length than global oxygen bias. After 3000 years and for a criterion of 50 mmol $O_2$m$^{-3}$ the OMZ volume bias varies between -31% and 54% of the observed volume of $56.7 \times 10^6$ km$^3$ (Figure 7). Especially the output of S6-All shows a very good agreement with observed OMZ volume, followed by the solution of S6$^*$-All and, surprisingly, of S4-Org (Figure 7) that still shows a declining trend (Figure 8).

Like global average oxygen the metric pattern of the model setups does not resemble any of the other metrics such as RMSE or global biogeochemical fluxes (compare Figure 7 with Figures 1 and 4). Simulated OMZ volume shows a very dynamic and non-linear trajectory, that depends strongly on the model configuration: for an OMZ defined by a criterion of $O_2 < 50$ mmol $O_2$m$^{-3}$ the simulated OMZ volume bias of individual model setups varies over time between 11% and 61% of the observed OMZ volume (Figure 8). All models show an initial decrease within the first few decades which is followed by an increase. The strength of the decline and the timing of the turning point seem to depend on $b$. Setup S4-Org with a high $b$ shows the most dynamic trajectory with a strong initial decline, a late turning point and a second turning point. This second

turning point is less pronounced in most other setups; it is absent in the solution of S4-SO that shows an almost continuous increase in OMZ volume bias.

After 3000 years the OMZ bias seems to depend mostly on the oxygen demand of remineralisation $R_{-O2:P}$. Models with the largest values of the parameter (S4-SO, S6-DOP) show a strong overestimate of OMZ volume, whereas L4-SO, with $R_{-O2:P} = 169.3$ mol $O_2$:mol P shows the most negative bias (Figures 7 and 8). The spread among the optimised models at

this time is almost as large (84%) as the observed volume of $56.7 \times 10^6$ km$^3$, but is still much smaller than the spread of $212.5 \times 10^6$ km$^3$ reported by Bopp et al. (2013).

Yet, even after 3000 years some models have not reached equilibrium with respect to the OMZ volume bias (Figure 8). Especially a large value of $b$ as in S4-Org induces a highly dynamic trajectory, which prevents a quantification of its final equilibrium state with regard to OMZ volume. The highly non-linear trajectories of the OMZ volume bias and the possible

presence of several turning points indicate that many processes play a considerable role for their evolution and these act on timescales of decades to at least centuries. Thus, any model skill assessment that relies on a continuous propagation of a trend simulated within the first few hundred years, such as applied by Seferian et al. (2016) for average oxygen, or implicitly assumed by Dietze and Loeptien (2013) might misrepresent the ultimate, equilibrium state of the model.

### 3.5 Contributions of tuning strategy and spin-up length to model uncertainty

As shown above, the optimised models exhibit a considerable spread among parametric model setups and also differ in $J_{\text{RMSE}}^{\text{post}}$ and biogeochemical fluxes and oxygen diagnostics when evaluated at two different time slices. Figure 9 illustrates the extent of variability arising from these two sources (parametric setup arising from optimisation vs. simulation time) for different diagnostics. For a given diagnostic the left and right boundary of a rectangle show the minimum and maximum range among the six different parametric model setups evaluated at two different time slices (after 10 and 3000 years of simulation). We

note that for most diagnostics the left boundary indicates the range among models when analysed after 10 years of spin-up, and the right boundary after 3000 years of spin-up. An exception to this is deep particle flux, where the variability among the models decreases over time (see above). The upper and lower boundary depict the maximum and minimum difference over time (10 and 3000 years) among the six individual model setups, i.e. the lower boundary indicates the value for model with the lowest temporal difference, and the upper boundary the value of the model with the largest difference. Hence, all

parametric and temporal differences of each diagnostic fall within the respective rectangle. A wide rectangle indicates that the variability caused by parametric setup is very different at the two different time slices. A high rectangle indicates that the temporal variability of model setups is very different for the individual parametric setups.

To obtain a common scale, the range of each diagnostic has been normalised by the respective average before evaluating the minimum and maximum. This approach allows to evaluate the potential variability (with regard to parametric setup and time)

of each diagnostic in comparison to other diagnostics (as indicated by the rectangles' position on the x- and y-axis). Likewise, it also allows to compare the variability induced by parametric setup (x-axis) to that of the spin-up length (y-axis): rectangles located in the lower right corner indicate diagnostics whose variability is dominated by parameters, whereas those in the upper

left corner are dominated by the spin-up length. For better visibility Figure 9 depicts the values on a logarithmic scale. The numerical values are also shown in Table S5.

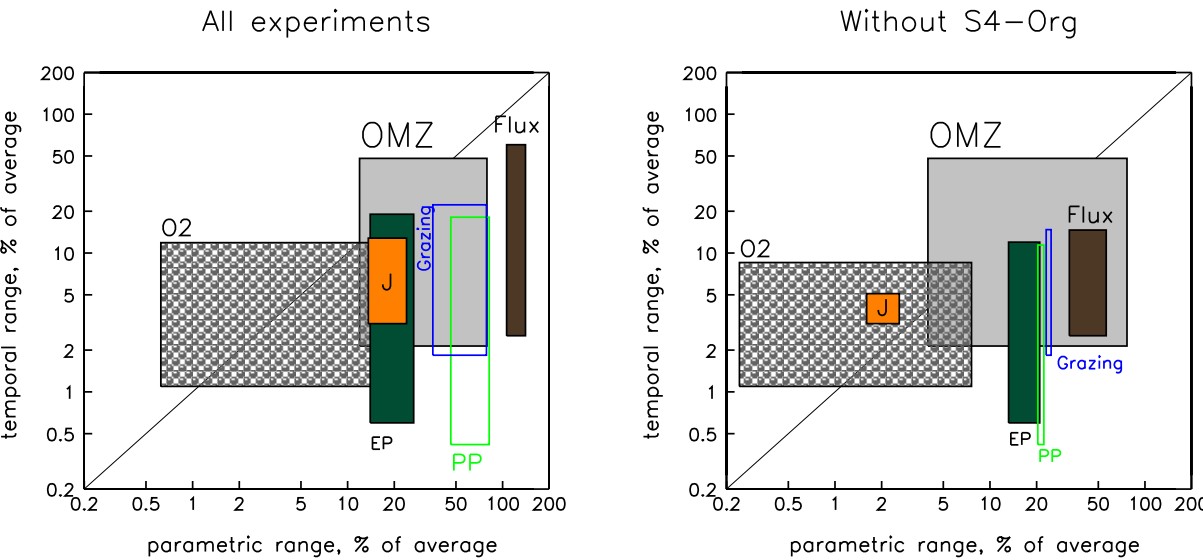

**Figure 9.** Graphic summarising the different sources of variability in model results, caused by model setup (parameter sets) and spin-up time before model analysis for biogeochemical fluxes, oxygen inventory and OMZ volume. The x-axis depicts the minimum and maximum variational range due to differences in model setup (parameters) after 10 and 3000 years of model spin-up time. The y-axis shows the minimum and maximum range due to spin-up time among the six individual model setups. All values have been normalised by the average over the respective axis (model set up or time), and are shown on a logarithmic scale. Rectangle colours or outlines denote the different diagnostics as labeled, with the orange rectangle indicating the range of $J_{\mathrm{RMSE}}^{\mathrm{post}}$ (equation 1). The region below the 1:1 diagonal indicates a higher variation because of the parametric setup, the region above a higher variation due to simulation time. Left panel: ranges across all six model setups, right panel: with S4-Org excluded. See Table S5 for numbers and details.

The spread between parametric model setups (see also $\Delta$ Parameters of Table S5) of primary production (green outlines in Figure 9) and grazing (blue outlines) is generally two- to three times larger than that of $J_{\mathrm{RMSE}}^{\mathrm{post}}$ (orange rectangle). Export production (dark green rectangle) shows only a small variation, of about the same size as $J_{\mathrm{RMSE}}^{\mathrm{post}}$, but its variability shows a declining trend over time (see Table S5). The quite narrow rectangle outlines of the biogeochemical fluxes indicate that inter-model variation persists more or less at the two time slices, i.e. after 10 and 3000 years of spin-up. Particle flux (dark
brown rectangle) exhibits the largest variation among the different parametric model setups. Because of the large differences in optimal $b$, and because this parameter controls the propagation of organic matter from the surface to the ocean interior (see above), deep particle flux after 10 years already varies by more than 140% of the average value (right border of dark brown rectangle and Table S5), which is more than 10 times the variation of $J_{\mathrm{RMSE}}^{\mathrm{post}}$. After 3000 years its variation is still four times that of $J_{\mathrm{RMSE}}^{\mathrm{post}}$. Because all models started from the same initial oxygen distribution, which does not change much within the
first decade of simulation, the variation among the different model setups with regard to oxygen (grey hatched rectangle) and

OMZ volume (grey rectangle) is quite small after 10 years ($< 1\%$ and $12\%$, respectively; see left borders of the corresponding rectangles and Table S5), but the variation of OMZ volume increases strongly with simulation time and reaches about $80\%$ of the model average after 3000 years (right border of grey rectangle and Table S5), which is comparable to the maximum variation of primary production and grazing.

The sensitivity to spin-up length (lower and upper borders of the rectangles in Figure 9) is very different for the individual model configurations, and can be very small, between $0.4\%$ for primary production (lower border of green rectangle) and $3.1\%$ for $J_{\mathrm{RMSE}}^{\mathrm{post}}$ (lower border of orange rectangle; see also $\Delta$ Time of Table S5). Individual models can exhibit a quite large variation over time. For instance, depending on parametric model setup the OMZ volume may change by up to $48\%$ (upper border of grey rectangle), and simulated global particle flux even by $60\%$ (upper border of brown rectangle), while primary production, grazing and export can only reach a maximum variation over time of about $20\%$ for individual model setups. However, much of the inter-model and temporal model spread is caused by the model setup S4-Org, that applies a high value for the particle flux parameter $b$.

As discussed above, the very large value of $b = 1.8$ in setup S4-Org induces slow sinking and shallow remineralisation, which, on long time scales increases subsurface nutrients (Figure S3), primary and export production, grazing and ultimately deep particle flux in the tropics and subtropics (Figure S2). These complex feedbacks result in a large temporal amplitude of biogeochemical fluxes on millennial time scales. Omitting S4-Org from the analysis (right panel in Figure 9) results in a general decrease in the spread of biogeochemical fluxes and average oxygen caused by different parameter settings (rectangles are shifted to the left), and also smaller differences in parametric setup at the two different time slices (rectangles become narrower). In particular, maximum variations due to model parametric setup are approximately only one half (after 10 years) to one third (after 3000 years) of the full ensemble for most biogeochemical fluxes (see also the values in parentheses of Table S5). The reduced ensemble also displays a lower maximum temporal variation for biogeochemical fluxes (the upper rectangle boundaries moving downwards). This is especially true for the maximum temporal variation of deep particle flux, whose maximum temporal change is reduced to less than $15\%$.

In contrast to the biogeochemical fluxes, the maximum variation in OMZ volume over time remains the same, because the reduced ensemble still includes setup S4-SO, which shows the largest amplitude of the model trajectory (Figure 8). To summarise, both model setup and spin-up time can play an important role for the simulation of primary production, grazing, deep particle flux, but especially for OMZ volume. Much of this variability disappears when omitting setup S4-Org, with its very high value for $b$ from the analysis; yet, the large sensitivity of OMZ volume to parameters and spin-up length remains, indicating that a variety of mechanisms and processes (such as the oxygen demand of remineralisation and large scale circulation) affect the evolution of this diagnostic.

An even larger variability exists for global models that also differ in circulation and/or model complexity ($\Delta$ GCMs of Table S5). Because global model spin-up times vary strongly (see also Figure 8), and because there is no general agreement on minimum spin-up times of global models (except for recent recommendations to spin-up unforced models for at least 500 to 2000 years; Eyring et al., 2016; Orr et al., 2017), it is difficult to disentangle the various sources of variability in these simulations. Our results suggest that mainly differences in biogeochemical model setup contribute to this, but a considerable

fraction can also be attributed to the spin-up length, especially for global OMZ volume, which is most sensitive to model parameters and spin-up length.

Our analysis of the different sources of model variability has so far focused on biogeochemical fluxes and oxygen. It remains to be investigated, if this analysis is applicable to tracers with different boundary exchanges and residence times, such as the carbon cycle, which is driven by an atmospheric signal and characterised by different residence times. Yet, at least after 250 years tracers related to the carbon system may still show a considerable drift that is in the range obtained for oxygen and nitrate, which warrants a careful consideration and potential correction when analysing and comparing model results (as, e.g., in Seferian et al., 2016).

## 4 Conclusions

We applied different optimisation strategies to calibrate a global biogeochemical ocean model against observed inorganic and organic tracers. Optimal parameter sets diverge strongly with regard to the particle flux exponent $b$, which is quite influential for the vertical and large-scale distribution of nutrients and oxygen (e.g. Kwon and Primeau, 2006; Kriest et al., 2012). The wide range of $b$ is mainly caused by an optimisation that disregarded observations of organic tracers, which resulted in a high optimal $b$ of 1.8. Optimisations that considered organic components arrived at $b \approx 1$ or less. Values of $b$ around one agree with results by Kwon and Primeau (2006), but are lower than those of earlier optimisations against inorganic tracers in the same circulation (Kriest et al., 2020). Likely, the low optimal values of $b$ found in the present study are caused by its (indirect) effect on phytoplankton: a low value of $b$ reduces nutrient recycling and hence primary production within the euphotic zone, which eventually leads to a less positive phytoplankton bias. We note, however, that especially $b$, but also other optimal model parameters not only depend on the data set and misfit applied for optimisation, but also on circulation (Kriest et al., 2020), especially when combined with long spin-up times. It remains to be investigated whether our results can be transferred to other model configurations that apply different circulations and/or model structures.

Despite the large range of some optimal model parameters, the resulting model solutions yield similar values of the normalised, volume-weighted root-mean-squared error ($J_{\text{RMSE}}^{\text{post}}$), showing a range $\leq 14\%$ of the average $J_{\text{RMSE}}^{\text{post}}$ after 10 years of simulation, and a range of 24% when extending the simulations with optimal parameters to 3000 years. Models calibrated with organic tracer data show some improved performance with regard to $J_{\text{RMSE}}^{\text{post}}$, mainly through a reduction in bias, and the difference in $J_{\text{RMSE}}^{\text{post}}$ decreases to less than 3%.

Since the root-mean-squared error combines bias and pattern error information, major improvements in model performance may not be well reflected by this metric during optimisation. For example, errors in circulation can cause a large pattern error, especially for sparse and episodic observations of organic tracers. These errors cannot be further reduced by any adjustment of biogeochemical parameter values. One way to examine the effects of circulation errors on the metric's ability to resolve optimal model parameters would be to calibrate the model against synthetic data derived from an earlier simulation (an "identical twin" experiment). Such an approach could provide deeper insight into the importance of organic tracers for model calibration and, if combined with different "sampling strategies" of the pseudo-data, also into the impact of data sparsity. However, the pseudo-

data will typically be representative for one particular model set-up, and may not reflect the full misfit sensitivity over a range of model characteristics. A second possible approach could involve the development and application of a an alternative misfit function. Such a metric could include an assessment of the tracers' observed and simulated statistical properties within specified ocean regions, instead of calculating local point-wise residuals as in the root-mean-squared error (e.g., Chien et al., 2022). We note that, in addition to the choice of data sets, a different choice in the mathematical form of the misfit function could considerably impact optimal parameter estimates and simulated biogeochemical turnover (Evans, 2003).

Diagnostics such as global primary production and grazing, global and local particle flux, oxygen bias and OMZ volume, show a large divergence among the models and over time, that is not reflected by differences in RMSE:

– Global primary production and grazing exhibit a similar patterns of model performance after 10 and 3000 years spin-up, but the difference between the model setups increases over the long spin-up, eventually becoming almost twice as high as the observational uncertainty of 40 Gt C y$^{-1}$. Yet, the range of primary production obtained in our study is well below the range documented for other global models, which can be attributed to differences in circulation and biogeochemical model complexity.

– Global export production is very similar among the optimal model setups, likely because all model runs presented here applied the same circulation, and because of the antagonistic effects of particle sinking and nutrient supply from subsurface waters. The variation of 14 to 17% found in our study is less than one fourth of the spread among other GCMs (which differ in many aspects, such as circulation), and much lower than the spread of observational estimates.

– Owing to the wide range of optimal estimates for parameter $b$, the variation among the different model configurations with regard to deep particle flux is larger than 100% of the average, and about as large as the spread across other GCMs and that of observed estimates. The temporal variation depends strongly on the applied $b$. With shallow remineralisation, as induced by a large value of $b = 1.8$, complex global feedback processes cause a considerable temporal variation of 60%, which reduces to less than 15% when this model setup is omitted. The differences in deep global (and local) particle flux between the individual model setups become smaller over time (in contrast to all other global biogeochemical fluxes, where the differences amplify over time). In addition, the setup with $b = 1.8$ that performs worst with regard to deep particle flux after 10 years also performs badly with regard to the global bias of oxygen and nitrate after 3000 years. Hence, the model's representation of particle flux might serve as an "early" criterion for model performance with regard to the potential long-term inventory of the non-conserved inorganic tracers, that do not depend on the initial conditions. It remains to be investigated if and to what extent deep particle flux can be used as a performance indicator when applied to other metrics, or when the model is started from initial conditions that differ from the observed climatologies applied in this study.

– Global average oxygen changes by up to 12% over time for the individual models, and the difference among the model setups is strongly amplified after 3000 years of simulation, when it reaches almost 22% of the ensemble mean. A large

fraction of this variation can be attributed to values of particle flux parameter $b$. For models with b $\leq$ 1, temporal variations of oxygen narrow down and differences of global average oxygen between model setups reduce to 8%.

– Global OMZ volume, when defined by $O_2 < 50$ mmol m$^{-3}$, varies by 48% of the ensemble mean after 3000 years. This variation is about one fifth of the spread across other global models that also vary with regard to physics, model complexity and spin-up time. We stress that simulated global OMZ volume is characterised by a highly non-linear trajectory over time with several turning points. This variation indicates that temporal extrapolations from some initial trend, as suggested by Seferian et al. (2016) and applied by Dietze and Loeptien (2013), do not provide robust estimates of the global OMZ volume.

The dependence of simulated global biogeochemical fluxes on tuning strategy and the resulting model parameters can have consequences for models of higher trophic levels such as fish, that often rely on primary production and/or export of organic matter to the mesopelagic and deep ocean. The interactions between biogeochemistry and fish may be further complicated through feedback effects of fish on biogeochemical features such as oxygen distributions (e.g., Bianchi et al., 2021), which can only be investigated through two-way coupled models (e.g., Aumont et al., 2018). A careful examination of model uncertainties with regard to simulated biogeochemical fluxes and OMZs, which might affect large, commercially relevant fish (Stramma et al., 2012), could support precautionary approaches to estimate present and future fish stocks (see also Schnute and Richards, 2001).

Overall, the best performance with regard to oxygen and OMZ volume was obtained by tuning strategies that either apply long (millennial) time scales of simulation, or that include observations of organic tracers in the misfit function. Given the computational expense of long-term simulations, and the likely dependence of global oxygen distribution on particle flux to the deep ocean, we speculate that well-confined observational estimates of particulate organic matter flux to the ocean interior may help to constrain global models, even when these are spun up only for a few decades or centuries.

*Code and data availability.* The basic TMM and MOPS code used for the ocean biogeochemical simulations are available to download from https://doi.org/10.5281/zenodo.1246300 (Khatiwala, 2018). Modifications to the MOPS code for the specific experiments described in this paper, along with model output, data sources and scripts to assemble the data sets used for optimisation are available under https://hdl.handle. net/20.500.12085/b174de1c-0bed-47f5-9718-7a8d44d1d2d1. The optimisation algorithm CMA-ES applied in this study is available from the Supplement of Kriest et al. (2017)

## Appendix A: Data sets and metrics

### A1   Equivalents to model tracers

**Nutrients and oxygen** Kriest et al. (2020) calibrated MOPS against observed nutrients and oxygen from interpolated climatologies (Garcia et al., 2006a, b). We here use the same data set, but restrict it to the upper 100 m, which reduces the

number of data points for each tracer by about one third (see Table A1). The restriction also approximately halves the (unweighted) global mean nutrient concentration, whereas average oxygen is slightly increased.

**Phytoplankton** For model calibration of phytoplankton we used chlorophyll data derived from remote sensing (MODIS-Aqua; Malin, 2013, downloaded on 08 April 2020). The surface data are available as a monthly climatology on a 9km grid. After averaging to annual mean chlorophyll, the data were averaged onto the ECCO grid. Chlorophyll was converted to carbon using the algorithm derived by Sathyendranath et al. (2009), and then to phosphorus using a C:P ratio of 122 mol C: mol P. The resulting data set contains 36.800 data points, which are all located in the surface layer (0-10m), with minimum and maximum values of 0 and 0.27 mmol P m$^{-3}$, respectively, and an unweighted mean of 0.016 mmol P m$^{-3}$ (see Table A1).

**Zooplankton** For model calibration of zooplankton we used the MAREDAT data set of mesozooplankton (Moriarty and O'Brien, 2013). This sparse data set contains 42.245 data points of monthly mean mesozooplankton (in mg C m$^{-3}$) on a $1 \times 1$ degree grid. After averaging over a year, and mapping onto the ECCO grid, we obtained a total of 35.202 data points. Conversion to phosphorus was carried out by assuming a C:P ratio of 122 mol C: mol P. The model does not distinguish between micro- and mesozooplankton, but aggregates both types into one single component. Unfortunately, observations of microzooplankton are much more sparse (only 2029 monthly data in the data set by Buitenhuis et al., 2013) than those of mesozooplankton, and often taken at other locations and during other times. Based on an analysis at stations where both small and large zooplankton observations are available, we estimated an approximate ratio of micro-to-mesozooplankton of one. For comparison with the model we therefore multiplied the data obtained from meso-zooplankton observations by two, resulting in minimum and maximum concentrations of 0 and 0.272 mmol P m$^{-3}$ and an unweighted average of 0.006 mmol P m$^{-3}$ (see Table A1). Restricting the gridded data to the upper 100 m reduces the sample size by about one third, with little effect on the global average.

**Particulate organic matter (detritus)** There is no direct observational equivalent to simulated detritus; the nearest type of observations are probably those of particulate organic phosphorus (POP), nitrogen (PON) or carbon (POC). Due to the methods applied, these observations, however, also contain phytoplankton and possibly a fraction of smaller zooplankton. For model evaluation we downloaded the data set by Martiny et al. (2014, data set CNP_data_DRYAD_edit_2.csv, downloaded on 16 April 2020), which contains more than 40.000 entries of particulate organic matter (POM) in units of phosphorus (POP), nitrogen (PON) and carbon (POC). After omitting entries where depth was not given, we obtained 6940 data entries for POP, and 46.705 data entries for PON. Because of the much higher data frequency for PON, we used this variable as further diagnostic, and converted it to POP using a stoichiometric ratio of 16 mol N:mol P, that is also applied internally by the model. For regridding onto model grid, we averaged all data that fall within a $1 \times 1$ degree area, with depth intervals as in ECCO, without any consideration of sampling date, thereby obtaining 6.513 data points, with minimum and maximum concentrations of 0 and 1.69 mmol P m$^{-3}$, respectively, and an unweighted average of 0.052 mmol P m$^{-3}$ (see Table A1).

**Table A1.** Observational data: number of observations over the full model domain and for the upper 100 m only, minimum and maximum concentration and average concentration over full and domain and upper 100 m. See section A for further details.

| Type | Number | | Min | Max | Average | | Source |
|------|--------|--------|-----|-----|---------|--------|--------|
| | Full | 0-100 m | | | Full | 0-100 m | |
| $PO_4$ | 682.604 | 218.610 | 0.02 | 3.9 | 1.6 | 0.76 | Garcia et al. (2006a) |
| $NO_3$ | 682.604 | 218.610 | 0 | 49.2 | 21.6 | 8.71 | Garcia et al. (2006a) |
| $O_2$ | 682.604 | 218.610 | 2.00 | 406.5 | 206.7 | 256.2 | Garcia et al. (2006b) |
| Phytoplankton | 36.800 | 36.800 | 0 | 0.27 | 0.016 | 0.016 | Malin (2013) |
| (Meso)Zooplankton | 35.202 | 25.613 | 0 | 0.27 | 0.006 | 0.006 | Moriarty and O'Brien (2013) |
| POP | 6.513 | 4.354 | 0 | 1.69 | 0.052 | 0.068 | Martiny et al. (2014) |
| DOP | 1.445 | 814 | 0 | 3.92 | 0.145 | 0.181 | Torres-Valdes et al. (2009) |
| | | | | | | | Moutin et al. (2008) |
| | | | | | | | Yoshimura et al. (2007) |
| | | | | | | | Landolfi et al. (2008) |

**DOP** Most observations of dissolved organic phosphorus (DOP) have been compiled by Angela Landolfi. They include data from cruises 36N, AMT10, AMT12, AMT14, AMT15, AMT16 and AMT17 (Torres-Valdes et al., 2009), the BIOSOPE cruise (Moutin et al., 2008), as well as published data from the North Atlantic (cruise D279, April-May 2004, Landolfi et al., 2008) and the Indian Ocean (cruise CD139, March-April 2002; Landolfi unpubl.). In the compilation we only included data with a positive (good) quality flag. We further included data read from Figure 02 of Yoshimura et al. (2007). Data were gridded onto a $1 \times 1$ degree grid, with the depth axis as defined in ECCO. After regridding we obtained 1445 data points, with minimum and maximum values of 0 and 3.92 mmol P m$^{-3}$, and an unweighted average of 0.145 mmol P m$^{-3}$. Restricting the domain to the upper 100 m reduces the sample size to 814 data points and increases the global average concentration by about one quarter (see Table A1).

## A2  Data sets for particle flux

The data set by Honjo et al. (2008, Table 3) consists of 152 data points of particle flux, derived from at least annual deployments of sediment traps between 382 and 8431 m depth. The data set by Lutz et al. (2007, Table 1) includes 245 data points of particle flux, derived from sediment traps between 140 and 5847 m depth, most of which were deployed at least one year, or a long enough time to reproduce the seasonal cycle. Thirdly, Mouw et al. (2016) provide an extensive data set of sediment traps, including also many data from short-term deployments. Restricting this to data over a deployment period of at least 360 days we obtained 369 data points between depths of 200 to 5847 m depth. (Note that, because of model topography, the final number of data points for model comparison is less.)

*Author contributions.* IK designed and carried out the experiments. AL provided the data for DOP. IK prepared the manuscript with contributions from all co-authors.

*Competing interests.* The authors declare that they have no conflict of interest.

*Acknowledgements.* This work is a contribution to BMBF joint project CO2Meso (FKZ 03F0876A) and to RESCUE (EU grant agreement ID 101056939). Parallel supercomputing resources have been provided by the North-German Supercomputing Alliance (HLRN). The authors
wish to acknowledge use of the Ferret program of NOAA's Pacific Marine Environmental Laboratory for analysis and graphics in this paper. We thank Jörg Schwinger and an anonymous reviewer for their very helpful and constructive comments on the manuscript.

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
