# Peer review of "Exploring the role of different data types and timescales for the quality of marine biogeochemical model calibration"

_Biogeosciences, 2023_

## Referee Comment (RC1)

**Review of "Exploring the role of different data types and timescales for the quality of marine biogeochemical model calibration" by Kriest et al., for Biogeosciences**

Kriest et al., set out to quantify the impact of using different observational constraints when calibrating predominantly surface-based parameters in a biogeochemical model. They also quantify the impact of nominally short or long spin-up times when doing these calibrations. This is achieved through a series of biogeochemical model experiments using the same ocean circulation model. The authors find that the inclusion of organic tracers such as dissolved organic phosphorus led to improved calibration in terms of computational time, from both faster calibration and through use of shorter spin-ups, and a reduced misfit to observations. The authors also provide a detailed examination of the impact of spin-up time on various tracers.

Overall, this is a really interesting and useful contribution. Approaches to spin-ups are very varied and this provides practically-relevant evidence for the biogeochemical model community that biogeochemical models can be reliably calibrated with relatively short spin-ups. The analysis also contributes valuable scientific understanding about the different behaviour of tracers in spin-ups. Generally, the experimental design and analysis is detailed and robust but would benefit from some additional clarifications concerning one particular experiment and the issue of misfit generated from the ocean circulation model. The authors have done an impressive job to tackle what is a large topic but the downside is a manuscript that can be hard to follow in places. I think some additional definition of the quantitative concepts up front would provide a useful reference for readers to help navigate the manuscript better.

**General Comments**

- *Design of L4-SO experiment*

The experiment design is very logical overall and isolates the various differences as best as possible but the last experiment (L4-SO) adds two changes that should be isolated and aren't: the 3000 spin-up time and the switch to using global tracers as constraints. How can you be sure that the results from this experiment solely represent the inclusion of deep tracer information rather than the longer spin-up? This experiment is used for a large part of the oxygen discussion so this needs addressing with an additional "L4-SO-Surface" experiment to isolate the longer spin-up time from the use of global tracers. If this additional experiment is not too different from the L4-SO experiment then I would be happy if the authors documented this briefly in the supplementary and kept the existing experimental set-up and manuscript text with a brief note stating it is not a problem.

I can see that the two factors are potentially inter-linked because the longer spin-up will primarily allow for the deep tracers to equilibrate so it may be that the impact of one is implicit in the other. It would be really interesting to know if just extending the spin-up time whilst constraining against the same surface observations would have a different outcome.

- *Influence of errors from ocean circulation and interpretation of calibration*

One of the challenges the authors identify is that the ocean circulation model contributes a large part of the misfit which cannot be improved by the biogeochemical model calibration. This introduces a potential issue that the inclusion of organic tracers in the calibration is accounting for some misfit due to the ocean circulation, i.e., overfitting the biogeochemical model, rather than representing the fidelity of the biogeochemical model. In particular, the interplay between the organic tracers and the particle flux exponent *b* relates to nutrient cycling in the upper ocean which is also strongly related to the ocean circulation model.

One way to remove the circulation error would be to calibrate against an existing model set-up, such as the ECCO* experiment, that acts as a set of pseudo-observations. Such an approach could completely demonstrate that the inclusion of organic tracers improves the calibration. However, I appreciate this is could be a big piece of work! I think the approach used in the manuscript is valid because it best relates to the practical reality of calibrating biogeochemical models. If the authors are able to explore this alternative option, even if just as a briefer

complimentary set of experiments, then it would help strengthen their key findings. Otherwise, some additional discussion of this potential issue would be helpful.

- *Clarification of quantitative concepts used*

The methods section describes the RMSE misfit function used in the study, However there are a number of other quantities used in the study (bias, bias-normalised RMSE etc…) that are not described which made it hard to keep track of how they all relate to the interpretation and to each other. For example, this is particularly the case when discussing pattern-matching statistics and Taylor diagrams later in Section 3.2 and beyond. It felt like the authors are introducing new concepts for interpreting the data in these sections which makes the text harder to follow. It would really help with fully understanding what the authors are showing and describing if they can expand the existing section on RMSE to include an overview of the additional quantities, particularly how they relate to each other (e.g., Jolliff et al., 2009) and their use in interpreting the model performance.

**Specific Comments**

Lines 49 - 50: how many studies is this out of interest? A brief list of the studies with some details on what observations are used would be a great resource for the community - perhaps this could be added to the supplementary if not too onerous for the authors!

Lines 65 - 70: it seems worth mentioning how equilibrium can be defined, such as some dXdt quantity that is smaller than a defined threshold?

Lines 65 - 70: spin-up time will also depend on the initial conditions used - do the models you consider here all initialise from observations? Also, the spin-up time for a tracer like PO4 will be different to other tracers that involve additional processes like gas exchange, e.g., DIC.

Lines 85 - 86: This statement is a little unclear without having read the rest of the manuscript. I'm not quite sure whether this is referring to the way the misfit function is set up to balance the different constraints or whether this is referring to the focus on the surface ocean (in which case, it would help to add a sentence of clarification as to this assumption).

Line 114: "ECCO*" lead me looking for a footnote! It would help to clarify this is an abbreviation.

Line 127: "half of the model's zooplankton" - is this literally 0.5 * biomass?

Lines 132 - 137: are all the observations compared as annual averages?

Line 154: I think that 3000 years is an appropriate time for the model to reach equilibrium given the transport matrix circulation but it would help to confirm this is the case.

Lines 233 - 235: Does the convergence occur faster simply because there are less parameters to optimise?

Lines 235 - 236: Is it the spin-up time or the deep tracer constraints that drive the faster convergence? (See general comments above)

Line 278: "that targets only at dissolved" doesn't read particularly right to me, possibly there's a typo or grammar issue?

Figure 2: It would help to have an additional marker legend for the constraints

Figure 4: It's notable that although the EP fluxes are all very similar across the experiments, the flux to 2000m varies considerably! This makes me wonder whether the experiments have very different regenerated (and preformed) tracer inventories? For example, the S4-Org experiment seems like it would have to have lower regenerated inventories if the export flux is similar but so

little is getting delivered to the oldest ocean depths. Could this evidence for an additional constraint in calibrating the models?

Figure 4: It would be possible to add a shaded area for model predictions in panel D using the transfer efficiency values at 1000m from CMIP6 runs in Wilson et al., (2022). This would require changing the reference depth from 2000m to 1000m but at least for the Henson observations this could be calculated from the published fit to SSTs?

Figure 8: The information shown on spin-up time here would be useful in the introduction!

Lines 467 - 470: The upper/lower left/right description seems the wrong way round to the figure? I may be wrong, I found Figure 9 generally quite a challenging figure to interpret.

Figure 9: Overall, this is a challenging figure to interpret! I think part of the reason for this is that you have parametric and temporal ranges as axes with ranges also depicted by the rectangles, which may be leading to me misreading the figure and related text?

Lines 489 - 491: is the smaller temporal variation related to the shallower b? Does the shallower remineralisation mean that more of the temporal variation in tracers is weighted towards the faster-to-equilibrate upper ocean rather than the slower-to-equilibrate deep ocean?

Lines 505 - 510: this answer is somewhat specific to the tracers explored in this study. DIC and alkalinity may have different responses for example. This caveat should be mentioned.

Lines 546 - 547: the suggestion of an "early-criterion" may depend on what the initial conditions of the spin-up are? Would this still be the case if you spin the model up from uniform initial conditions? A clarification about initial conditions would be useful to make throughout the manuscript generally.

**References**

Jolliff et al., (2009) Summary diagrams for coupled hydrodynamic-ecosystem model skill assessment. Journal of Marine Systems. 76 (1-2), pp. 64 - 82

Wilson et al., (2022) The biological carbon pump in CMIP6 models: 21st century trends and uncertainties. PNAS. 119 (29)

---

## Author Comment (AC1)

We thank the referee for his / her encouraging and constructive comments! We address the suggestions and questions (highlighted in blue) in detail below (our answers in black).

Kriest et al., set out to quantify the impact of using different observational constraints when calibrating predominantly surface-based parameters in a biogeochemical model. They also quantify the impact of nominally short or long spin-up times when doing these calibrations. This is achieved through a series of biogeochemical model experiments using the same ocean circulation model. The authors find that the inclusion of organic tracers such as dissolved organic phosphorus led to improved calibration in terms of computational time, from both faster calibration and through use of shorter spin-ups, and a reduced misfit to observations. The authors also provide a detailed examination of the impact of spin-up time on various tracers.

Overall, this is a really interesting and useful contribution. Approaches to spin-ups are very varied and this provides practically-relevant evidence for the biogeochemical model community that biogeochemical models can be reliably calibrated with relatively short spin-ups. The analysis also contributes valuable scientific understanding about the different behaviour of tracers in spin-ups. Generally, the experimental design and analysis is detailed and robust but would benefit from some additional clarifications concerning one particular experiment and the issue of misfit generated from the ocean circulation model. The authors have done an impressive job to tackle what is a large topic but the downside is a manuscript that can be hard to follow in places. I think some additional definition of the quantitative concepts up front would provide a useful reference for readers to help navigate the manuscript better.

General Comments

- Design of L4-SO experiment

The experiment design is very logical overall and isolates the various differences as best as possible but the last experiment (L4-SO) adds two changes that should be isolated and aren't: the 3000 spin-up time and the switch to using global tracers as constraints. How can you be sure that the results from this experiment solely represent the inclusion of deep tracer information rather than the longer spin-up? This experiment is used for a large part of the oxygen discussion so this needs addressing with an additional "L4-SO-Surface" experiment to isolate the longer spin-up time from the use of global tracers. If this additional experiment is not too different from the L4- SO experiment then I would be happy if the authors documented this briefly in the supplementary and kept the existing experimental set-up and manuscript text with a brief note stating it is not a problem.

I can see that the two factors are potentially inter-linked because the longer spin-up will primarily allow for the deep tracers to equilibrate so it may be that the impact of one is implicit in the other. It would be really interesting to know if just extending the spin-up time whilst constraining against the same surface observations would have a different outcome.

Indeed, optimisation L4-SO combines two changes to the overall misfit at the same time, namely the spin-up time and the consideration of deep inorganic tracers. For the latter to yield useful results, the longer spin-up time is essential. Thus, as correctly noted by the reviewer, the two aspects are inter-linked. A short spin-up time with deep tracers included the misfit would reflect mainly the initial conditions of deep inorganic tracers. Thus, deep tracer concentrations only become informative after a long (enough) spin-up time.

The reviewer's suggestion is to investigate whether the extended spin-up time introduces different results compared to S4-SO. According to our results we readily find solutions for the surface ocean

after short periods that agree with the solution after a much longer spin-up time, although deep inorganic tracers were regarded in one case. As shown in the discussion paper, the a posteriori evaluation of the (surface) misfit after 3000 years results in almost the same misfit values and its components as those after 10 years (see, for example, the upper right panel of Fig. 1). Likewise, biogeochemical fluxes (Figure 4) and other metrics (Figure 2) of L4-SO are very similar at the two time slices. Hence, we do not expect an optimisation against surface tracers after a spin up time of 3000 years to result in a fundamentally different set of parameters or biogeochemical turnover. Given this potential insensitivity of the surface misfit and other metrics to length of spin-up time, we have refrained from carrying out such a computationally expensive optimisation. **We will comment on this in the discussion of a revised version of the paper.**

- Influence of errors from ocean circulation and interpretation of calibration

One of the challenges the authors identify is that the ocean circulation model contributes a large part of the misfit which cannot be improved by the biogeochemical model calibration. This introduces a potential issue that the inclusion of organic tracers in the calibration is accounting for some misfit due to the ocean circulation, i.e., overfitting the biogeochemical model, rather than representing the fidelity of the biogeochemical model. In particular, the interplay between the organic tracers and the particle flux exponent b relates to nutrient cycling in the upper ocean which is also strongly related to the ocean circulation model.

One way to remove the circulation error would be to calibrate against an existing model set-up, such as the ECCO* experiment, that acts as a set of pseudo-observations. Such an approach could completely demonstrate that the inclusion of organic tracers improves the calibration. However, I appreciate this is could be a big piece of work! I think the approach used in the manuscript is valid because it best relates to the practical reality of calibrating biogeochemical models. If the authors are able to explore this alternative option, even if just as a briefer complimentary set of experiments, then it would help strengthen their key findings. Otherwise, some additional discussion of this potential issue would be helpful.

We thank the reviewer for this very constructive and helpful suggestion. Indeed, an identical twin experiment with pseudo-data or respective subsets (with different subsampling strategies) would be a great opportunity to explore this issue further. So far, and within the time frame available for our response, we are not able to devise such an experimental setup for additional optimisations, but we would like to follow up on this in the future. We note that the pseudo-data is typically representative for one particular model solution, which means that we should now have sufficient prior knowledge about the model behaviour to come up with a meaningful twin. **We will comment on the effect of circulation and the potential future directions in a revised version of the manuscript.**

- Clarification of quantitative concepts used

The methods section describes the RMSE misfit function used in the study, However there are a number of other quantities used in the study (bias, bias-normalised RMSE etc...) that are not described which made it hard to keep track of how they all relate to the interpretation and to each other. For example, this is particularly the case when discussing pattern-matching statistics and Taylor diagrams later in Section 3.2 and beyond. It felt like the authors are introducing new concepts for interpreting the data in these sections which makes the text harder to follow. It would really help with fully understanding what the authors are showing and describing if they can expand the existing section on RMSE to include an overview of the additional quantities, particularly how they relate to each other (e.g., Jolliff et al., 2009) and their use in interpreting the model performance.

We agree that the presentation and discussion of the different metrics (RMSE, normalised RMSE, unbiased RMSE, bias, Pearson correlation coefficient) can be confusing. The individual metrics looked at are essentially all part of the RMSE and normalised RMSE. There are no additional quantities. How the individual parts relate to each other has been nicely addressed in Jolliff et al. (2009) and in Taylor (2001). **We suggest adding a brief, more formal description in the Methods section, which also refers to the previous works of Jolliff et al. (2009) and Taylor (2001).**

Specific Comments

Lines 49 - 50: how many studies is this out of interest? A brief list of the studies with some details on what observations are used would be a great resource for the community - perhaps this could be added to the supplementary if not too onerous for the authors!

The sentence in lines 49-50 relates to the statement by Arhonditsis & Brett (2004): *"Less than 5% of the modeling studies assessed included information (statistics or time-series plots) for all the state variables predicted, thus we were not able to evaluate overall model performance."* We discussed this reviewer's comment and realised that the reference to the work of Arhonditsis and Brett (2004), which included local, regional and large scale models, may not be representative for the current state of model assessments, almost twenty years after their study was published. To highlight this, we will rephrase the sentence as **"Almost two decades ago, far less than half of the studies reviewed by Arhonditsis and Brett (2004) reported performance statistics for all simulated state variables."**

We agree that a comprehensive and updated overview on model-data comparison would be very informative and helpful. Coming up with some detailed update is not straightforward and would be a study of its own, even if restricted to global biogeochemical model applications. For example, while the initial presentation and evaluation of global BGC models often focuses on model skill only with regard to inorganic tracers, often subsequent studies look more deeply into organic components (e.g., Gehlen et al., 2006; Petrik et al., 2022). We will mention that the situation with regard to model evaluation seems to be improving (for CMIP5 and CMIP6 models). We will also add the range of zooplankton metrics from the study by Petrik et al. (2022) in the revised manuscript (lines 305ff).

Lines 65 - 70: it seems worth mentioning how equilibrium can be defined, such as some dXdt quantity that is smaller than a defined threshold?

There may be different criteria to define equilibrium, or near steady state, e.g. by an Euclidean norm (e.g., Priess et al., 2013). In general, as different quantities (phosphorus, nitrogen, carbon …) are of different magnitude, one may rather define the requirement of a maximum relative change for each tracer over the steady annual cycle (as suggested by the reviewer). **We will explicitly mention and present the term "equilibrated" in the introduction.**

Lines 65 - 70: spin-up time will also depend on the initial conditions used - do the models you consider here all initialise from observations? Also, the spin-up time for a tracer like PO4 will be different to other tracers that involve additional processes like gas exchange, e.g., DIC.

The inorganic tracers are initialised from observed distributions, and the organic tracers from globally homogenous concentrations of 0.0001 mmol P/m3. **We will clarify this in Section 2.4.**

So far, our model does not include any DIC, but only phosphate, nitrate and oxygen. Indeed the spin-up times for oxygen and nitrate (for example) can be quite different: because nitrate concentration and its global inventory are affected by the spatial distribution of, and distances

between, denitrification and nitrogen fixation.. In the model predominant regions for denitrification and nitrogen fixation are the OMZ in the eastern tropical Pacific and the subtropics in the Atlantic and Pacific respectively. Because these distant regions of fixed nitrogen loss and gain are connected through circulation, the nitrate inventory equilibrates only after several millennia. Oxygen, in contrast, adjusts on shorter time scales, depending on the biogeochemical model parameter values (see also Kriest and Oschlies, 2015, Fig. 2). **We will comment on the different transient behaviour of different tracers, and their dependence on biogeochemical parameters in the introduction.**

Lines 85 - 86: This statement is a little unclear without having read the rest of the manuscript. I'm not quite sure whether this is referring to the way the misfit function is set up to balance the different constraints or whether this is referring to the focus on the surface ocean (in which case, it would help to add a sentence of clarification as to this assumption).

We agree, the statement could be misinterpreted. The statement should relate to the "calibration bias", in reference to the work by Arhonditsis and Brett (2004). For example, changes in model parameters could lead to a reduction in surface phosphate bias, at the cost of a larger bias in dissolved organic phosphorus (Kriest, 2017). The bias in DOP will, however, remain overlooked if we excluded thus tracer from the misfit analysis (or optimisation). **We will rephrase this more clearly in a revised version of the paper.**

Line 114: "ECCO*" lead me looking for a footnote! It would help to clarify this is an abbreviation.

The name "ECCO*" relates to the name in Kriest et al. (2020). **We will clarify this in the revision of the paper.**

Line 127: "half of the model's zooplankton" - is this literally 0.5 * biomass?

Yes, we will exchange the current phrase by **"half of the zooplankton's biomass".**

Lines 132 - 137: are all the observations compared as annual averages?

Yes, we will change this to **"simulated and observed annual mean tracers"**. We will also **add a sentence that clarifies that we neglect any temporal variation.**

Line 154: I think that 3000 years is an appropriate time for the model to reach equilibrium given the transport matrix circulation but it would help to confirm this is the case.

As noted above, 3000 years may even be too short - this depends indeed on the biogeochemical constants applied (see above). A perfect optimisation setup would derive the spin-up time depending on, e.g., a Euclidean norm; however, this so far does not seem feasible in the current set-up of optimisation, where 10 model simulations with different sets of parameters run in parallel - here we may end up with very different simulation times  (depending on parameter value) when aiming at a common criterion for steady state, which could result in a large potential computational overhead. **We will add a comment on this in a revised version of the revision.**

Lines 233 - 235: Does the convergence occur faster simply because there are less parameters to optimise?

In general, the convergence depends on the characteristic of the parameter-cost/misfit function manifold and on how the optimisation algorithm copes with it. Faster convergence can be achieved with a reduction of the dimensionality of the problem (smaller number of parameters), by the introduction of additional (informative) data constraints, or by both combined. Since in this

particular case (S6-All vs S6-DOP) we only reduced the number of parameters to be optimised, we assume that the faster convergence indeed arises because of this reduction in the dimensionality of the problem. **To clarify this, we will add a comment in a revised version of the paper.**

Lines 235 - 236: Is it the spin-up time or the deep tracer constraints that drive the faster convergence? (See general comments above)

The distribution of deeper tracer concentrations results from the combination of biogeochemical processes in conjunction with ocean circulation processes. Because the latter processes include millennial timescales, effects on deep tracers on the misfit are only informative after long spin-up times. We therefore cannot provide a conclusive answer to this question, as already explained above, in the reply to the general comments. However, we do know that large-scale mismatches in deep tracer concentrations, which only occur after a long enough spin-up time, are highly informative for model parameters such as $b$ (e.g., Kwon et al., 2006, Kriest et al., 2012, Kriest et al., 2017).

Line 278: "that targets only at dissolved" doesn't read particularly right to me, possibly there's a typo or grammar issue?

**We will replace "targets only at" by "considers only the misfit to".**

Figure 2: It would help to have an additional marker legend for the constraints

**We will provide a marker legend for the constraints (tracer types).**

Figure 4: It's notable that although the EP fluxes are all very similar across the experiments, the flux to 2000m varies considerably! This makes me wonder whether the experiments have very different regenerated (and preformed) tracer inventories? For example, the S4-Org experiment seems like it would have to have lower regenerated inventories if the export flux is similar but so little is getting delivered to the oldest ocean depths. Could this evidence for an additional constraint in calibrating the models?

We agree, an optimisation against a preformed and/or regenerated observational counterpart would be very useful. We so far have not implemented such "synthetic" tracers directly into the model. It would be possible to derive this information from AOU and stoichiometric assumptions - but, in cases where these assumptions (e.g., the value of $R_{O2:P}$) are simultaneously affected during the course of optimisation, the situation may become complicated: for example, should we apply respective values of the parameter $R_{O2:P}$ (that is itself subject to optimisation) for deriving regenerated nutrients from observations? Also, the circulation error of the models may become even more important, as it will affect the saturation concentration of oxygen that enters the calculation of AOU. Overall, the reviewer's comment is inspiring for interesting and more elaborate approaches to optimisation.

We would like to note that in S4-Org because of the very large $b$ (=very shallow remineralisation) a large fraction of the EP will be recycled in the upper water column (e.g., about 30 % of EP will be recycled between 100 and 200 m). As EP is typically largest in the productive high latitudes (with deep winter mixing) or in upwelling areas (see Fig S2) this fraction of EP will likely be returned to the euphotic zone on seasonal time scales. Hence, when relying on annual tracer concentrations for model assessment, the definition of regenerated and preformed may be somewhat ambiguous (in contrast to the flux in 2000 m).

Figure 4: It would be possible to add a shaded area for model predictions in panel D using the transfer efficiency values at 1000m from CMIP6 runs in Wilson et al., (2022). This would require

changing the reference depth from 2000m to 1000m but at least for the Henson observations this could be calculated from the published fit to SSTs?

We thank the referee to point to the publication by Wilson et al. (2022), but we would prefer to keep the reference depth of Fig. 4 at 2000 m, as many observational data sets refer to that particular depth. **We will additionally compare our accomplished (diagnosed from flux at 1000m divided by EP) transport efficiency TE with those reported in Wilson et al. (2022).** Wilson et al. report a global mean TE between 0.03 (UKESM1-0-LL) and 0.25 (IPSL-CM5A2-INCA), and a global observed TE of about 20 %. The full range of our model setups includes values as low as 6 % (S4-Org) up to 23 % (S6*-All), which agrees with the range reported by Wilson et al. (2022). Our model simulations with $b=1$, after a spin-up of 10 years exhibit a global mean TE of 0.17-0.18, which agrees with the observed range. Furthermore, we would like to note that there is a discrepancy between TE diagnosed from simulated fluxes and the nominal TE that can be calculated directly from $b$, via $(100 \text{ m}/1000 \text{ m})^b$. For example, $b=1$ results in a TE of 0.1. This discrepancy arises from many facts, for example the (additional) transport of particulate organic matter through mixing, as well as numerical diffusion (Kriest and Oschlies, 2011). This should be kept in mind when parameterising and analysing the models.

Figure 8: The information shown on spin-up time here would be useful in the introduction!

**We will extend and detail our presentation on model spin-up times in the introduction by the range shown in Figure 8.**

Lines 467 - 470: The upper/lower left/right description seems the wrong way round to the figure? I may be wrong, I found Figure 9 generally quite a challenging figure to interpret.

**Yes, we agree, this figure is challenging, and we will try to describe this figure in a more comprehensible way.** Figure 9 condenses and illustrates the outcome of our cross-validation experiment (i.e., variation among parametric setups vs variation due to the time at which the models are evaluated), and serves as a graphic illustration of the results depicted in detail in Table S4. Typically (except for particle flux at 2000 m) the left boundary of each rectangle shows the spread of model diagnostics after 10 years of simulation, and the right boundary after 3000 years. The lower and upper boundaries shows the difference between 10 and 3000 years, with the lower boundary indicating the model with the minimum difference (of the entire ensemble of six model setups) and the upper boundary the model with the maximum difference. Hence, all parametric and temporal differences fall within each rectangle.

Figure 9: Overall, this is a challenging figure to interpret! I think part of the reason for this is that you have parametric and temporal ranges as axes with ranges also depicted by the rectangles, which may be leading to me misreading the figure and related text?

See above: the rectangles illustrates the domain of uncertainty (or variation) due to spin-up time and model parameters. **We will try to explain this figure and its interpretation better. In particular, the first paragraph of 3.5 will be revised, to clarify the link between Figure 9 and the results of the cross-validation experiments.**

Lines 489 - 491: is the smaller temporal variation related to the shallower b? Does the shallower remineralisation mean that more of the temporal variation in tracers is weighted towards the faster-to-equilibrate upper ocean rather than the slower-to-equilibrate deep ocean?

As can be seen from Figure 4, biogeochemical fluxes of S4-Org (the model configuration with the largest $b$ or shallowest remineralisation) shows the largest differences between year 10 and year 3000. This can be explained with the complex feedbacks that occur at a global scale, where for

example, a "shallow" $b$ eventually (on long timescales) increases subsurface nutrients (Figure S3), primary and production, grazing and ultimately deep particle flux in the tropics and subtropics (Figure S2). Removing this member from the ensemble leads to a lower maximum temporal variation of most biogeochemical fluxes (compare upper boundaries of left and right panels of Figure 9). Hence, the very high $b$ of S4-Org triggers the largest temporal variation, whereas low temporal variation is achieved by models with $b=1$ and less. For OMZ volume, S4-SO and S6-DOP show the largest temporal variation (Figure 8). This likely arises from the complex processes (physical and biogeochemical) that determine OMZ extent, in conjunction with the high oxygen demand of remineralisation ($R_{O2:P}$ = 200 mol $O_2$:mol P) and $b=1$. So ultimately, the large temporal effects can be traced back to large-scale (remote) effects on long time scales that redistribute subsurface nutrients. **To clarify this, we will add a sentence in the following paragraph (currently line 493), that emphasises this again.**

Lines 505 - 510: this answer is somewhat specific to the tracers explored in this study. DIC and alkalinity may have different responses for example. This caveat should be mentioned.

We agree, the spin-up times of DIC and alkalinity may be quite different, as also indicated by Seferian et al. (2013). **We will comment on this in the revised version.**

Lines 546 - 547: the suggestion of an "early-criterion" may depend on what the initial conditions of the spin-up are? Would this still be the case if you spin the model up from uniform initial conditions? A clarification about initial conditions would be useful to make throughout the manuscript generally.

**We agree, and will restrict this statement to models that are started from observed inorganic tracer distributions.**

References

Jolliff et al., (2009) Summary diagrams for coupled hydrodynamic-ecosystem model skill assessment. Journal of Marine Systems. 76 (1-2), pp. 64 - 82

Wilson et al., (2022) The biological carbon pump in CMIP6 models: 21st century trends and uncertainties. PNAS. 119 (29)

Arhonditsis and Brett (2004) Evaluation of the current state of mechanistic aquatic biogeochemical modeling. Mar. Ecol., Prog. Ser., 271, doi:10.3354/meps271013

Kriest and Oschlies (2011) Numerical effects on organic matter sedimentation and remineralization in biogeochemical ocean models. Ocean Modell., 39, doi:10.1016/j.ocemod.2011.05.001

Kriest et al. (2012) Sensitivity analysis of simple global marine biogeochemical models. Glob. Biogeochem. Cy., 26, GB2029, doi:10.1029/2011GB004072

Kriest and Oschlies (2015) MOPS-1.0: towards a model for the regulation of the global oceanic nitrogen budget by marine biogeochemical processes. Geosci. Mod. Dev., 8, doi:10.5194/gmd-8-2929-2015

Kriest et al. (2017) Calibrating a global three-dimensional biogeochemical ocean model (MOPS-1.0). Geosci. Mod. Dev., 10, doi:10.5194/gmd-10-127-2017

Kriest et al. (2020) One size fits all? Calibrating an ocean biogeochemistry model for different circulations. Biogeosciences, 17(12), doi:10.5194/bg-17-3057-2020

Kwon and Primeau (2006) Optimization and sensitivity study of a biogeochemistry ocean model using an implicit solver and in situ phosphate data. Glob. Biogeochem. Cy., 20, GB4009, doi:10.1029/2005GB002631

Leles et al. (2016) Evaluation of the complexity and performance of marine planktonic trophic models. Annals of the Brazilian Academy of Sciences, 88, doi:10.1590/0001-3765201620150588

Moriarty & O'Brian (2013) Global distributions of mesozooplankton abundance and biomass - Gridded data product (NetCDF) - Contribution to the MAREDAT World Ocean Atlas of Plankton Functional Types. Earth System Science Data, 5, doi:10.5194/essd-5-45-2013

Petrik et al. (2022) Assessment and Constraint of Mesozooplankton in CMIP6 Earth System Models. Glob. Biogeochem. Cy., 36(11), e2022GB007367, doi:10.1029/2022GB007367

Priess et al. (2013) Accelerated parameter identification in a 3D marine biogeochemical model using surrogate-based optimization. Ocean Modell., 68, doi:10.1016/j.ocemod.2013.04.003

Seferian et al. (2013) Inconsistent strategies to spin up models in CMIP5: implications for ocean biogeochemical model performance assessment. Geosci. Mod, Dev., 9, doi:10.5194/gmd-9-1827-2016

Taylor (2001) Summarizing multiple aspects of model performance in a single diagram. J. Geophys. Res., 1006, doi:10.1029/2000JD900719

---

## Author Comment (AC2)

We thank the referee Jörg Schwinger for his insightful and very constructive comments! Our replies to his questions, comments, and proposed changes (highlighted in blue) are given below (our answers in black).

The authors present results from an extensive parameter optimization study for a global scale ocean biogeochemistry model. They evaluate the impact of different optimisation set-ups, in terms of observation based data used, parameters to be optimized, and model spin-up length. Particularly, they focus on the benefit that assimilation of organic tracer information (phyto- and zooplankton, POP and DOP) might have for the optimization of ocean biogeochemical models. The authors generally find that the inclusion of organic tracer data has a strong impact on the representation of particle sinking in the model and improves model fidelity with respect to global oxygen biases and the represenation of oxygen minimum zones. Different spin-up times are shown to have a considerable influence on global oxygen and OMZ volume biase, even for optimized parameter sets that perform similarly with respect to surface nutrient and oxygen data.

Parameter otimization of global biogeochemical models is understudied due to its complexity and technical challenges, and this study is a very welcome addition to the field. The results are useful (beyond the technical aspects of optimization) for a wide community of ocean biogechemical and Earth system modellers since they tell us about model sensitivity in general. The paper is well written and the method is sound. I didn't find any problem with this manuscript, except a few rather minor points where the manuscript would benefit from clarifications. These points along with technical corrections are listed below.

General:

1) It is a bit confusing that the term J_RSME (equation 1) is used to denote the misfit (cost) function used in the optimization procedure, but also for the a-posteori (after optimization) quantification of model misfit. Although both misfit functions take the same form (eq. 1), they are different in which data and regions are considered. This is not explained in the methods section, rather there are only some hints scattered in the text. For example, the reader can guess what is shown in Fig. 1 based on the fact that the J_RMSE values are the same as in table 2, where there is a note in the caption. To make this more transparent, I would suggest to describe this two-fold use of the misfit function in the methods section. I also would prefer to use a (slighly) different notation for the cost function and the a-posteori misfit function (e.g. \hat{J} for the cost function).

We agree that the distinction between the misfit function applied during optimisation and the one used for a posteriori evaluation should be described more clearly. **We will extend the Methods section related to the misfit function to clarify this, and also include more description on the other metrics applied throughout this paper (see comments by Rev. 1), such as (unweighted) RMSE, unbiased RMSE, bias and Pearson correlation correlation coefficient.**

2) The representativeness of the data products used for optimisation is discussed several places in the manuscript, but it would be helpful to gather a short description of this aspect in the methods section. The WOA climatologies for nutrients and oxygen should provide to first order a like-to-like comparison with the coarse model and the climatological ocean circulation. The same is probably true for the chlorophyl data? The discussions that are found later in the manuscript point towards the fact that the Martiny data is too sparse to be representative of the simulated model counterpart. It would then also be useful to frame the later discussions more consistently as a problem of representativeness, for example lines 262-266: Is this really a problem of the coarse resolution, or more the problem that the available data are not representative of a climatological average over a 1x1 degree gridcell? The same comment applies for lines 286-287, and also for lines 321-328 (where representativeness is finally mentioned).

Also, is it really plausible that the lack of correlation (for large scale global patterns) can be explained by errors in the (data assimilated) ocean circulation? Doesn't this potentially also point towards a too low model complexity, i.e. only one phytoplankton and zooplankton type?

We  suggest addressing the potential problems that arise from sparsity and episodic nature of the organic observations (except phytoplankton) in the Methods section (2.2) more extensively. This sparsity and episodicity, in conjunction with the global model setup (in short: a coarse, climatological circulation), and the neglect of temporal variability in the misfit function may introduce the following causes for the lack in pattern matching: (1) There can be a mismatch between the climatological circulation in comparison to the hydrographic conditions that prevailed during sampling of the observations; (2) The observations, which usually provide only snapshots of the biogeochemical state, may not be representative for the annual average that  we apply in our misfit function; (3) the biogeochemical complexity may be too low (i.e. the model is too simple).

(1) and (2) are somewhat intertwined, and could be addressed with a physical model that more realistically resolves the physical environment at high spatial and temporal scales. However, it would be difficult to run such a model at a global scale, and over long time scale. Even if we had a model that resolves eddies and filaments well, it may still be possible that an eddy would occur at a slightly different location (i.e., a few kilometers further north), or with a temporal delay. In this case, any misfit function that aims at an exact spatial and temporal match would create a large error. One possible solution would be to use difference metrics such as the Hellinger distance to assess model performance, and we briefly discuss this at the end of section 3.2. (3) Too low model complexity is addressed briefly in the discussion about data-model comparison with regard to zooplankton, where at least for large zooplankton quite comprehensive direct (Moriarty and O'Brian, 2013) and indirect (e.g., Petrik et al., 2022) data sets exist. The situation is worse for microzooplankton, a common component in many global models, and for other model components such as DOP and POP,  where data sets are indeed very sparse. Hence, even with a more complex model we may have to face the same problem of data sparsity in time and space.

**To address these issues more concisely and comprehensively, we suggest to skip the occasional references to the potential causes of RMSE, RMSE' and correlation mismatch in lines 262-266, 286-287 and elsewhere, and rather combine these in a more exhaustive discussion at the end of this section, with reference to the (sparse) data coverage presented more extensively in section 2.2**

More specific comments:

-line 2: "...state of the ocean biogeochemistry..." I would find it justified to delete the word biogeochemistry here. The models tell us something about the ocean in general.

**We will delete this.**

-line 15-16: "mainly located in surface layers" is a bit unclear, please consider rewording.

**We will replace this sentence by : When evaluating the RMSE of tracers located in the upper 0-100 m (except for particulate organic matter, for which we consider the entire vertical domain) we find similar values for the different model setups, with a range of 14% …"**

-line 32: Consider adding "combined with data assimilation techniques" or similar after "Global biogeochemical ocean models"

**We will add this: "Global biogeochemical ocean models, especially when combined with data assimilation techniques, …"**

-line 49: "Far less than half of the studies...". Unclear which studies this refers to. Please consider rewording to clarify this.

The sentence in lines 49-50 relates to the statement by Arhonditsis & Brett (2004): *"Less than 5% of the modeling studies assessed included information (statistics or time-series plots) for all the state variables predicted, thus we were not able to evaluate overall model performance."* **To highlight also the fact that this study is almost two decades old we will rephrase the sentence as "Almost two decades ago, far less than half of the studies reviewed by Arhonditsis and Brett (2004) reported performance statistics for all simulated state variables." We will also briefly comment on a few recent advance in model assessment for global CMIP5/CMIP6 models.** (See also our reply to comment by Rev 1.)

-line 82: "one of the simulated compartments." It is unclear to me what "compartments" refers to (inorganic/organic? tracers/fluxes? nitrogen/phosphourous?)

This refers to all biogeochemical state variables simulated by the model. **We will rephrase this by "at least one of the biogeochemical state variables simulated by the model."**

-line 83: "basic optimisation procedure". Is "basic" a good word here? Maybe better "reference"?

We would prefer to reserve the word "reference" to the simulation ECCO* by Kriest et al. (2020), **but will change "basic" to "initial".**

-line 79-90: It is confusing that it reads "three further experiments" and "these five optimisations". The fact that the "basic" optimisation is actualy two different optimisations (one with range of b more constrained) is difficult to understand. Please consider explaining this better.

**We will mention the fact that the initial optimisation setup against the full data set is carried out with two different boundary ranges for *b*, and then highlight that all further optimisations consider a reduced data set.**

-line 101: consider changing "a circulation" to "a circulation field"

**We will change this.**

-line 170: instead of saying "the DOP parameters", the two parameters could be spelled out for clarity.

**We will spell out the two DOP parameters.**

-line 169-174: I don't understand the logic behind this experiment: If the objective is "to analyse whether the neglect of iron limitation in MOPS yields a bias in parameter estimates" then why is the number of paramters to be optimized changed at the same time? This way it is not clear whether changes in the optimized model performance are due to the change in data coverage or due to different set of parameters to be optimized? Could the authors please comment on this?

In fact, the (short-term) optimisations presented in this paper are only a subset of a larger ensemble of optimisations. We have also carried out an experiment similar to S4-SO, where we include the phytoplankton data in the Southern Ocean in the misfit (as in S6-DOP), but only optimise four model parameters (as in S4-SO). The results of this optimisation (the four optimal model parameters) were very similar to to those of S4-SO, and resulted in an optimal Ic = 28.45 W/m2/d (compared to 28.84 of S4-SO), and optimal zooplankton grazing rate of 2.967 1/d (compared to 2.807 of S4-SO), a stoichiometry of 199.5 (compared to 200 of S4-SO) and a *b* of 1 (the same as in

S4-SO). We note that these values lie well within the 0.1% range of misfit for S4-SO. To keep the plots and analysis simple, we have not added this optimisation in the tables and figures. **However, as this optimisation may provide valuable information and clarify the (small) impact of Southern Ocean data, we suggest to mention these results briefly in the Methods section as follows (lines 174ff):**

**"We note that in an intermediate step between S6-DOP and S4-SO we carried out an optimisation similar to S4-SO, but with the Southern Ocean data included. This optimisation resulted in parameters which were very similar to those obtained from S4-SO, and hence produced very similar model results. We therefore do not include these in our model analysis."**

If regarded helpful, we can also provide an extended version of Tables 1 and 2 in a supplement, where we indicate the setup (Table 1) and misfit, iterations and optimal parameters (and their ranges) for this optimisation. We would, however, prefer to not include its results in the figures and analysis of the main part, as this may reduce the visibility of the plots and does not seem to add to the overall insights and conclusions.

-Table 1: instead of using italic font for fixed parameters, why not write "fixed at 0" and "fixed at 0.1848" for the two cases where this is relevant? Would be easier in my oppinion.

**Yes, we agree and will follow this suggestion.**

-line 204: "introduced" sounds odd to me. Maybe just "used"?

**We will change this.**

-line 231: Why "a likely larger number"? L is given in the table, or does this refer to something else? Please clarify.

**We are sorry for this confusion: "Likely" survived from an earlier version of this manuscript, and we will delete the word.**

-line 252-253: "...in contrast to a reduction of JRMSE by about 25% for the consideration of DOP measurements." I don't understand this. In Fig 1a I don't see a very significant difference between S6-all and S6-DOP?

We are sorry that this was expressed ambiguously. What we meant to say was that JRMSE of DOP improved by 25% (Table S1), and **we will rephrase the sentence by "in contrast to a reduction of JRMSE of DOP by 25% (Table S2)."** As "particulate organic tracer concentrations" can be ambiguous (one may think of POP, which, in fact, deteriorates slightly compared to ECCO*), **we will also replace this term by "plankton concentrations".**

-Figure 4, caption: For panel b (grazing), I can't see any short or long horizontal bar?

**We are sorry for this confusion: The sentence relating to short and long horizontal bars relates to an earlier version of this figure, and we will delete the sentence.**

-Figure 7, caption: "Numbers on top of the panels..." I can't see any numbers on top of panels?

**We are sorry for this confusion: When preparing the final version of this Figure we skipped the numbers (as these are already in Figure 8). We will delete the sentence.**

-Figure 9, caption: I don't think "sources of variability" is a good wording, please consider rewording. Also, "which is different for the individual model setups": It is actually only the L4-SO setup that is different, right? So maybe "which is different for L4-SO compared to the other setups"

"Sources of variability": **We suggest to replace this by "sources of variability in model results"**. "which is different for the individual model setups": We agree, this expression is not clear. We here do not want to refer to the spin-up times applied to the different optimisations, but to the spin-up times after which each model was analysed (a posteriori). **We will replace this expression by "due to the spin-up time, after which each model was analysed" and skip the phrase in parentheses.**

Technical/typos:

-line 28: non -> not

**We will change this.**

-line 273: expose -> show (?)

**We will change this.**

-line 278: organics -> organic tracer data

**We will change this.**

-line 370: agrees with -> falls within

**We will change this.**

-line 373: too low -> below all other estimates

**We will change this.**

References:

Arhonditsis and Brett (2004) Evaluation of the current state of mechanistic aquatic biogeochemical modeling. Mar. Ecol., Prog. Ser., 271, doi:10.3354/meps271013

Kriest et al. (2020) One size fits all? Calibrating an ocean biogeochemistry model for different circulations. Biogeosciences, 17(12), doi:10.5194/bg-17-3057-2020

Moriarty & O'Brian (2013) Global distributions of mesozooplankton abundance and biomass - Gridded data product (NetCDF) - Contribution to the MAREDAT World Ocean Atlas of Plankton Functional Types. Earth System Science Data, 5, doi:10.5194/essd-5-45-2013

Petrik et al. (2022) Assessment and Constraint of Mesozooplankton in CMIP6 Earth System Models. Glob. Biogeochem. Cy., 36(11), e2022GB007367, doi:10.1029/2022GB007367

---

## Author Response (AR1)

We here briefly list a brief point-by-point reply (in black) as to how we have responded in the revised manuscript to the reviewers comments (blue). A more detailed reply is given in the discussion forum. Line numbers refer to the markup file submitted with this list, that highlights the changes made. Beside our changes made in response to the reviewers we have also spotted an error in table 2, a wrong reference, and modified a sentence to be more specific. These are listed at the end of this file.

**Reviewer 1:**

- Design of L4-SO experiment

The experiment design is very logical overall and isolates the various differences as best as possible but the last experiment (L4-SO) adds two changes that should be isolated and aren't: the 3000 spin-up time and the switch to using global tracers as constraints. How can you be sure that the results from this experiment solely represent the inclusion of deep tracer information rather than the longer spin-up? This experiment is used for a large part of the oxygen discussion so this needs addressing with an additional "L4-SO-Surface" experiment to isolate the longer spin-up time from the use of global tracers. If this additional experiment is not too different from the L4- SO experiment then I would be happy if the authors documented this briefly in the supplementary and kept the existing experimental set-up and manuscript text with a brief note stating it is not a problem. I can see that the two factors are potentially inter-linked because the longer spin-up will primarily allow for the deep tracers to equilibrate so it may be that the impact of one is implicit in the other. It would be really interesting to know if just extending the spin-up time whilst constraining against the same surface observations would have a different outcome.

We now discuss the potential effect of the combination of these two changes (length of spin-up time and consideration of deep tracers) in section 3.2, in particular lines 360-372 in the markup file.

- Influence of errors from ocean circulation and interpretation of calibration

One of the challenges the authors identify is that the ocean circulation model contributes a large part of the misfit which cannot be improved by the biogeochemical model calibration. This introduces a potential issue that the inclusion of organic tracers in the calibration is accounting for some misfit due to the ocean circulation, i.e., overfitting the biogeochemical model, rather than representing the fidelity of the biogeochemical model. In particular, the interplay between the organic tracers and the particle flux exponent b relates to nutrient cycling in the upper ocean which is also strongly related to the ocean circulation model. One way to remove the circulation error would be to calibrate against an existing model set-up, such as the ECCO* experiment, that acts as a set of pseudo-observations. Such an approach could completely demonstrate that the inclusion of organic tracers improves the calibration. However, I appreciate this is could be a big piece of work! I think the approach used in the manuscript is valid because it best relates to the practical reality of calibrating biogeochemical models. If the authors are able to explore this alternative option, even if just as a briefer complimentary set of experiments, then it would help strengthen their key findings. Otherwise, some additional discussion of this potential issue would be helpful.

We now discuss the potential effects of circulation, and the possible solution via a twin experiment in section 4 (Conclusions), in lines 688-694 in the markup file.

- Clarification of quantitative concepts used

The methods section describes the RMSE misfit function used in the study, However there are a number of other quantities used in the study (bias, bias-normalised RMSE etc...) that are not

described which made it hard to keep track of how they all relate to the interpretation and to each other. For example, this is particularly the case when discussing pattern-matching statistics and Taylor diagrams later in Section 3.2 and beyond. It felt like the authors are introducing new concepts for interpreting the data in these sections which makes the text harder to follow. It would really help with fully understanding what the authors are showing and describing if they can expand the existing section on RMSE to include an overview of the additional quantities, particularly how they relate to each other (e.g., Jolliff et al., 2009) and their use in interpreting the model performance.

We have extended section 2.3 (Misfit function) by explaining different metrics and their potential relation more clearly, with references to Taylor (2001) and Jolliff et al., (2009). We also refer throughout the paper to the different metric concepts as presented in 2.3, and distinguish more clearly between the misfit function applied during optimisation ($J_{RMSE}^{opt}$) and the metric evaluated a posteriori ($J_{RMSE}^{post}$).

Lines 49 - 50: how many studies is this out of interest? A brief list of the studies with some details on what observations are used would be a great resource for the community - perhaps this could be added to the supplementary if not too onerous for the authors!

As stated in our reply, we agree that a comprehensive and updated overview on model-data comparison would be very informative and helpful, but would probably be a study of its own, even if restricted to global biogeochemical model applications. However, we now added some examples of more comprehensive model assessment studies (such as the one by Petrik et al., 2022) and reworded our reference to the study by Arhonditsis & Brett (2004) to be more specific (see lines 51-65 in the markup file).

Lines 65 - 70: it seems worth mentioning how equilibrium can be defined, such as some dXdt quantity that is smaller than a defined threshold?

We now comment on the concept and criteria for equilibrium in the introduction (lines 74-100 of the markup file) and also address our applied spin up times in section 2.4 (lines 224-230).

Lines 65 - 70: spin-up time will also depend on the initial conditions used - do the models you consider here all initialise from observations? Also, the spin-up time for a tracer like PO4 will be different to other tracers that involve additional processes like gas exchange, e.g., DIC.

We have will clarified  this in Section 2.4 (lines 215-216), and comment on spin up in the introduction and in lines 224-230 in the markup file.

Lines 85 - 86: This statement is a little unclear without having read the rest of the manuscript. I'm not quite sure whether this is referring to the way the misfit function is set up to balance the different constraints or whether this is referring to the focus on the surface ocean (in which case, it would help to add a sentence of clarification as to this assumption).

See above, we have reworded and clarified out statement regarding the reference to the study by Arhonditsis & Brett (2004). At the same time we have skipped the reference to Leles et al. (lines 51-64 in the markup file).

Line 114: "ECCO*" lead me looking for a footnote! It would help to clarify this is an abbreviation.

We try to clarify this by hyphens around the name (line 144 in the markup file)**.**

Line 127: "half of the model's zooplankton" - is this literally 0.5 * biomass?

We changed the phrase to "half of the zooplankton's biomass" (line 159 in markup file).

Lines 132 - 137: are all the observations compared as annual averages?

We now describe more explicitly that we refer to the annual means in lines 178-180, and comment on the reasoning and possible effects of this assumption in the last paragraph of section 2.2 (lines 165-174 in markup file).

Line 154: I think that 3000 years is an appropriate time for the model to reach equilibrium given the transport matrix circulation but it would help to confirm this is the case.

We now comment on this in section 2.4, lines 224-230.

Lines 233 - 235: Does the convergence occur faster simply because there are less parameters to optimise?

We comment on the reason for faster convergence in lines 315-316.

Lines 235 - 236: Is it the spin-up time or the deep tracer constraints that drive the faster convergence? (See general comments above)

See above, additionally we discuss the potential effect of the combination of these two changes (length of spin up time and consideration of deep tracers) now in section 3.2, in particular in lines 360-372.

Line 278: "that targets only at dissolved" doesn't read particularly right to me, possibly there's a typo or grammar issue?

We replaced "targets only at" by "considers only the misfit to" (line 375).

Figure 2: It would help to have an additional marker legend for the constraints

We now provide a marker legend for the constraints (tracer types).

Figure 4: It's notable that although the EP fluxes are all very similar across the experiments, the flux to 2000m varies considerably! This makes me wonder whether the experiments have very different regenerated (and preformed) tracer inventories? For example, the S4-Org experiment seems like it would have to have lower regenerated inventories if the export flux is similar but so little is getting delivered to the oldest ocean depths. Could this evidence for an additional constraint in calibrating the models?

We do not comment in the revised manuscript on the extensive topic of regenerated vs preformed nutrients, but more discuss the feedback effect by shallow remineralisation in S4-Org, for example in the revised section 3.5.

Figure 4: It would be possible to add a shaded area for model predictions in panel D using the transfer efficiency values at 1000m from CMIP6 runs in Wilson et al., (2022). This would require changing the reference depth from 2000m to 1000m but at least for the Henson observations this could be calculated from the published fit to SSTs?

As stated in our detailed reply to reviewer 1 we would prefer to keep the reference depth of Fig. 4 at 2000 m. In lines 482-497 we compare our accomplished transport efficiency TE (diagnosed from flux at 1000m divided by EP) with the ones reported in Wilson et al. (2022).

Figure 8: The information shown on spin-up time here would be useful in the introduction!

We extended our presentation on model spin-up times in the introduction, including a reference to the source of the range (Seferian et al., 2020) shown in Figure 8.

Lines 467 - 470: The upper/lower left/right description seems the wrong way round to the figure? I may be wrong, I found Figure 9 generally quite a challenging figure to interpret.

We have tried to describe this figure in the caption and text in a more comprehensible way (section 3.5).

Figure 9: Overall, this is a challenging figure to interpret! I think part of the reason for this is that you have parametric and temporal ranges as axes with ranges also depicted by the rectangles, which may be leading to me misreading the figure and related text?

See above: We have tried to describe this figure in the caption and text in a more comprehensible way (section 3.5).

Lines 489 - 491: is the smaller temporal variation related to the shallower b? Does the shallower remineralisation mean that more of the temporal variation in tracers is weighted towards the faster-to-equilibrate upper ocean rather than the slower-to-equilibrate deep ocean?

In the revised section 3.5 we now try no express more clearly that a large temporal variation arises from shallow remineralisation because of complex, large scale feedbacks on longer time scales.

Lines 505 - 510: this answer is somewhat specific to the tracers explored in this study. DIC and alkalinity may have different responses for example. This caveat should be mentioned.

We now note this caveat in lines 661-666 .

Lines 546 - 547: the suggestion of an "early-criterion" may depend on what the initial conditions of the spin-up are? Would this still be the case if you spin the model up from uniform initial conditions? A clarification about initial conditions would be useful to make throughout the manuscript generally.

We have clarified the initial conditions in section 2.4, and now comment more explicitly on the interplay between the model's fit to observed global and local particle flux and the inventory of non-conserved tracers such as oxygen or nitrate, as well as the initial conditions in general.

**Reviewer 2:**

1) It is a bit confusing that the term J_RSME (equation 1) is used to denote the misfit (cost) function used in the optimization procedure, but also for the a-posteori (after optimization) quantification of model misfit. Although both misfit functions take the same form (eq. 1), they are different in which data and regions are considered. This is not explained in the methods section, rather there are only some hints scattered in the text. For example, the reader can guess what is shown in Fig. 1 based on the fact that the J_RMSE values are the same as in table 2, where there is a note in the caption. To make this more transparent, I would suggest to describe this two-fold use of the misfit function in the methods section. I also would prefer to use a (slighly) different notation for the cost function and the a-posteori misfit function (e.g. \hat{J} for the cost function).

We now extended the methods section (2.3) and included more description of the other metrics applied throughout this paper (see comments by Reviewer 1). These descriptions now include the RMSE, unbiased RMSE, bias and Pearson correlation coefficient. We also refer throughout the

paper to the different metric concepts as presented in 2.3, and distinguish more clearly between the misfit function applied during optimisation ($J_{RMSE}^{opt}$) and the metric evaluated a posteriori ($J_{RMSE}^{post}$).

2) The representativeness of the data products used for optimisation is discussed several places in the manuscript, but it would be helpful to gather a short description of this aspect in the methods section. The WOA climatologies for nutrients and oxygen should provide to first order a like-to-like comparison with the coarse model and the climatological ocean circulation. The same is probably true for the chlorophyl data? The discussions that are found later in the manuscript point towards the fact that the Martiny data is too sparse to be representative of the simulated model counterpart. It would then also be useful to frame the later discussions more consistently as a problem of representativeness, for example lines 262-266: Is this really a problem of the coarse resolution, or more the problem that the available data are not representative of a climatological average over a 1x1 degree gridcell? The same comment applies for lines 286-287, and also for lines 321-328 (where representativeness is finally mentioned). Also, is it really plausible that the lack of correlation (for large scale global patterns) can be explained by errors in the (data assimilated) ocean circulation? Doesn't this potentially also point towards a too low model complexity, i.e. only one phytoplankton and zooplankton type?

We skipped the occasional references in section 3.2 to the potential causes of large RMSE and RMSE' as well as low correlation. Instead we extended the presentation of data sparsity in the Methods section (2.2) and discuss the lack of improvement in pattern metrics at the end of section 3.2 (lines 417-426 in the markup file).

-line 2: "...state of the ocean biogeochemistry..." I would find it justified to delete the word biogeochemistry here. The models tell us something about the ocean in general.

We have deleted this (line 2 in markup file)

-line 15-16: "mainly located in surface layers" is a bit unclear, please consider rewording.

We replaced this sentence by: "Following the optimisation procedure we evaluated the RMSE for all tracers located in the upper 100 m (except for POP, for which we considered the entire vertical domain), regardless of their consideration during optimisation. " (lines 16-18 in markup file).

-line 32: Consider adding "combined with data assimilation techniques" or similar after "Global biogeochemical ocean models"

We rephrased this to: "Global biogeochemical ocean models, especially when combined with data assimilation techniques, …" (line 34 in markup file).

-line 49: "Far less than half of the studies...". Unclear which studies this refers to. Please consider rewording to clarify this.

We have rephrased parts of this paragraph to clarify this. We now also highlight the fact that the study by Arhonditsis and Brett is almost two decades old, and that the situation may have improved. (lines 51-64; see also our reply to comment by Rev 1.)

-line 82: "one of the simulated compartments." It is unclear to me what "compartments" refers to (inorganic/organic? tracers/fluxes? nitrogen/phosphourous?)

We have replaced this by "at least one of the biogeochemical state variables simulated by the model." (line 107 in markup file).

-line 83: "basic optimisation procedure". Is "basic" a good word here? Maybe better "reference"?

We have replaced "basic" by "initial" (line 107)

-line 79-90: It is confusing that it reads "three further experiments" and "these five optimisations". The fact that the "basic" optimisation is actualy two different optimisations (one with range of b more constrained) is difficult to understand. Please consider explaining this better.

We now mention that the initial optimisation setup against the full data set is carried out with two different boundary ranges for $b$, and then highlight that all further optimisations consider a reduced data set (see lines 107-117).

-line 101: consider changing "a circulation" to "a circulation field"

We have changed this (line 131).

-line 170: instead of saying "the DOP parameters", the two parameters could be spelled out for clarity.

We now spell out the two DOP parameters in the descriptions of the experiments.

-line 169-174: I don't understand the logic behind this experiment: If the objective is "to analyse whether the neglect of iron limitation in MOPS yields a bias in parameter estimates" then why is the number of paramters to be optimized changed at the same time? This way it is not clear whether changes in the optimized model performance are due to the change in data coverage or due to different set of parameters to be optimized? Could the authors please comment on this?

As noted in our detailed reply, we have also carried out an optimisation with all phytoplankton data included, but decided not to show this additional experiment, which yielded results similar to those of S4-SO. We mention this fact in the description of S4-SO (lines 246-255), and also provide an extended version of Table 2 in the supplement, where we also indicate the misfit, iterations and optimal parameters (and their ranges) for this additional optimisation.

-Table 1: instead of using italic font for fixed parameters, why not write "fixed at 0" and "fixed at 0.1848" for the two cases where this is relevant? Would be easier in my oppinion.

We have changed this.

-line 204: "introduced" sounds odd to me. Maybe just "used"?

We have changed this. (line 285)

-line 231: Why "a likely larger number"? L is given in the table, or does this refer to something else? Please clarify.

We deleted the word. (line 312)

-line 252-253: "...in contrast to a reduction of JRMSE by about 25% for the consideration of DOP measurements." I don't understand this. In Fig 1a I don't see a very significant difference between S6-all and S6-DOP?

We rephrased the sentence by "in contrast to a reduction of $J_{RMSE}^{post}$ of DOP by 25% (Table S2)." (line 335)

-Figure 4, caption: For panel b (grazing), I can't see any short or long horizontal bar?

We deleted the sentence.

-Figure 7, caption: "Numbers on top of the panels..." I can't see any numbers on top of panels?

We deleted the sentence.

-Figure 9, caption: I don't think "sources of variability" is a good wording, please consider rewording. Also, "which is different for the individual model setups": It is actually only the L4-SO setup that is different, right? So maybe "which is different for L4-SO compared to the other setups"

We replaced this by "sources of variability in model results, caused by model setup (parameter sets) and spin-up time before model analysis (…)". This comes along with a hopefully better description of this figure in the caption as well as in the text.

-line 28: non -> not

We have changed this. (line 30)

-line 273: expose -> show (?)

We have changed this. (line 356)

-line 278: organics -> organic tracer data

We have changed this. (line 374)

-line 370: agrees with -> falls within

We have changed this. (line 476)

-line 373: too low -> below all other estimates

We have changed this. (line 479)

**Additional changes:**

Table 2: The upper value for the good range of $I_c$ of S4-SO is 31.8 instead of 39.9 (as wrongly given in the discussion paper). This has now been corrected.

The reference to Seferian et al. (2013) in lines 69, 71, 76, 462 and 556 of the discussion paper was wrong, and should have been a reference to Seferian et al. (2016). This has now been corrected. (See lines 91, 95, 100, 585 and 731 in the markup file).

We have reordered the sentence in line 555-557 (conclusions) of the discussion paper to clarify that we are referring to the extrapolation of trends in general, not to that of global OMZ volume in particular (lines 731-732 in the markup file).

The DOP data by Landolfi listed as "unpubl." in the discussion paper are now referenced to Landolfi et al. (2008).